METHODS AND RESOURCES

# Completing the BASEL phage collection to unlock hidden diversity for systematic exploration of phage–host interactions

**Dorentina Humolli[1ʘ], Damien Piel[1ʘ], Enea Maffei[1,2‡], Yannik Heyer[2‡], Elia Agustoni[2‡], Aisylu Shaidullina[1,2], Luc Willi[2], Patrick Imwinkelried[2], Fabienne Estermann[2], Aline Cuénod[3], Dominik P. Buser[2], Carola Alampi[4], Mohamed Chami[4], Adrian Egli[3¤], Sebastian Hiller[2], Matthew Dunne[5], Alexander Harms** [iD][1,2*]

**1** Institute of Food, Nutrition, and Health (IFNH), ETH Zürich, Zürich, Switzerland, **2** Biozentrum, University of Basel, Basel, Switzerland, **3** Division of Clinical Bacteriology and Mycology, University Hospital Basel, Basel, Switzerland, **4** BioEM Lab, Biozentrum, University of Basel, Basel, Switzerland, **5** Micreos GmbH, Wädenswil, Switzerland

ʘ These authors contributed equally to this work.
‡ EM, YH, and EA also contributed equally to this work.
¤ Current address: Institute for Medical Microbiology, University of Zürich, Zürich, Switzerland
* alexander.harms@hest.ethz.ch

## Abstract

Research on bacteriophages, the viruses infecting bacteria, has fueled the development of modern molecular biology and inspired their therapeutic application to combat bacterial multidrug resistance. However, most work has so far focused on a few model phages which impedes direct applications of these findings in clinics and suggests that a vast potential of powerful molecular biology has remained untapped. We have therefore recently composed the BASEL collection of *Escherichia coli* phages (BActeriophage SElection for your Laboratory), which made a relevant diversity of phages infecting the *E. coli* K-12 laboratory strain accessible to the community. These phages are widely used, but their assorted diversity has remained limited by the *E. coli* K-12 host. We have therefore now genetically overcome the two major limitations of *E. coli* K-12, its lack of O-antigen glycans and the presence of resident bacterial immunity. Restoring O-antigen expression resulted in the isolation of diverse additional viral groups like *Kagunavirus*, *Nonanavirus*, *Gordonclarkvirinae*, and *Gamaleyavirus*, while eliminating all known antiviral defenses of *E. coli* K-12 additionally enabled us to isolate phages of *Wifcevirus* genus. Even though some of these viral groups appear to be common in nature, no phages from any of them had previously been isolated using *E. coli* laboratory strains, and they had thus remained largely understudied. Overall, 37 new phage isolates have been added to complete the BASEL collection. These phages were deeply characterized genomically and phenotypically with regard to host receptors, sensitivity to antiviral defense systems, and host range. Our results highlighted dominant roles of the O-antigen barrier for viral host recognition and of restriction-modification systems in bacterial immunity. We anticipate that the completed BASEL collection will propel research on phage–host interactions and their molecular mechanisms, deepening our understanding of viral ecology and fostering innovations in biotechnology and antimicrobial therapy.

**Data availability statement:** All data generated or analyzed during this study are included in this published article. The raw data of every replicate experiment underlying the plots shown in Figs 5D, 6G, 7D, 8G, 9E, 10E, 11A, 11B, S1A and S1B, all calculations underlying the summary data in these figures, and additional analyses are compiled in S1 Data. Tree files of all phylogenies shown in this work and the underlying sequence alignments in NEXUS format are available in S3 Data file. The code used to run GAPS for the results presented in this work is available on Github (https://github.com/hiller-lab/gaps) and on Zenodo (https://www.doi.org/10.5281/zenodo.14277981). The annotated genome sequences of all 37 newly isolated phages as well as Lederbergvirus isolates Dewey, Huey, and Louie (Figs 10D and S6B) are available in NCBI BioProject PRJNA1207239 with GenBank accession numbers listed in S2 Table.

**Funding:** This work was supported by the Swiss National Science Foundation (SNSF; https://www.snf.ch/en) Ambizione Fellowship PZ00P3_180085 (to AH), SNSF Starting Grant TMSGI3_211369 (to AH), and SNSF National Centre of Competence in Research (NCCR) AntiResist (to AH; grant number 180541). The funder played no role in the study design, data collection and analysis, decision to publish, or preparation of the manuscript.

**Competing interests:** The authors have declared that no competing interests exist.

**Abbreviations:** CGSC, Coli Genetic Stock Center; EOP, efficiency of plating; LB, Lysogeny Broth; NGR, N4 glycan receptor; PBS, Phosphate-buffered saline; PDB, Protein Data Bank; RBDs, receptor-binding domains; RBPs, receptor-binding proteins; RM, restriction-modification; ST, sequence type.

## Introduction

Bacteriophages, the viruses infecting bacteria, have attracted great interest due to their abundance and diversity in all ecosystems, their key roles as inspiration for biotechnology, and their therapeutic potential in the fight against resilient bacterial infections [1–4]. Most cultured phages are so-called tailed phages or *Caudoviricetes* that characteristically use a tail structure at the virion to inject their genome into host cells. Morphologically, tailed phages can be classified as myoviruses (contractile tail), siphoviruses (long, flexible, noncontractile tail), and podoviruses (short stubby tail). Following the "phage treaty" declared by Max Delbrück in 1944 [5], a few tailed phages infecting *Escherichia coli* have become key model systems for studying the molecular biology of phage–host interactions [6]. We therefore know comparably much about the molecular mechanisms underlying infections of *E. coli* with these seven "T phages" T1–T7 [7,8] and a few additional ones infecting *E. coli* or other major model organisms like *Bacillus subtilis*. While this work has resulted in many major breakthroughs since the "golden era" of molecular biology [3,9], the limited number and diversity of these model phages is a severe limitation for systematic research that could inform the effective use of natural phage diversity for therapeutic or biotechnological applications. We and others have therefore proposed broader sets of model phages infecting commonly studied host organisms such as *E. coli* [10–12], *Klebsiella* [13], or *Pseudomonas aeruginosa* and other pseudomonads [14–16]. Our previous work had presented 68 well-characterized new *E. coli* phages as the BASEL (BActeriophage SElection for your Laboratory) collection which we have made accessible to the academic community.

The value of these BASEL phages lies in their "assorted diversity," providing access to a broad spectrum of biological diversity both within and between phage groups when exploring any aspect of phage–host interactions [6,10]. For example, the BASEL collection has been used to study phage host recognition [17], viral dependence on host codons [18], porin functionality [19], phage replication rates [20], and bacterial immunity [21–25]. A key prerequisite for broad applicability of the BASEL phages is that they have deliberately been isolated using the *E. coli* K-12 laboratory strain as host, the most important model organism in fundamental microbiology research [9,26]. However, phage isolation with *E. coli* K-12 as host has also conflicted with the goal of assorted phage diversity by introducing inherent biases that prevented us from sampling a complete overview of *E. coli* phage diversity [10]. Firstly, the domesticated *E. coli* K-12 laboratory strain has lost the O-antigen glycan chains on the lipopolysaccharide (LPS) core that form a dense barrier on wild *E. coli* and shield cell surface receptors from viral recognition [27–30]. While the absence of this barrier made it easier to isolate multiple phages from different viral groups, it naturally excluded the isolation of phages that depend on the O-antigen as host receptor [31]. Furthermore, a second level of sampling bias may have been caused by known (and possibly unknown) antiviral defenses encoded by *E. coli* K-12, although we had deleted some of them already in the host strain used to isolate and study the original BASEL collection [10].

In the current study, we have addressed both limitations of our previous work by expanding the BASEL collection with 37 new phages that largely depend on (restored) O-antigen expression and of which some are inhibited by resident antiviral defenses in the K-12 genome in the form of O-antigen glycosylation. Notably, while almost none of these phages belong to groups of which other representatives have previously been found to infect *E. coli* K-12, all these phages do infect *E. coli* K-12 with a few easily implemented modifications and can thus be used for systematic fundamental research as an expansion of the original BASEL collection. Similar to our previous study, we have performed a systematic phenotypic and genomic characterization of these phages with a view to host receptors, bacterial immunity, and host range, and we make them accessible to the scientific community. These results extend some of

our previous observations and complement them with new discoveries on previously under-studied phage groups such as the *Gordonclarkvirinae* podoviruses that are abundant in nature, highly resistant to tested phage-defense systems, and have a comparably broad host range. The dependence of many newly included phages on O-antigen as host receptor highlights the dominant role of viral adsorption for *E. coli* phage host range and provides a valuable tool for studying diverse mechanisms of glycan recognition using the well-studied model strain *E. coli* K-12. Taken together, our findings provide novel insights into the ecology and evolution of bacteriophages as well as into their arms race with each other and bacterial immunity. The now completed BASEL collection comprises 106 phages that have been isolated from diverse sources and belong to more than 30 genera (S1 and S2 Tables). We anticipate that our work will serve as a reference point for future studies in fundamental phage biology and strengthen the BASEL collection as an effective and widely used tool in the scientific community.

## Results

### Overview: Including O-antigen-dependent phages in the BASEL collection

The cell surface of Gram-negative bacteria is dominated by LPS and proteins embedded in the outer membrane on which additional layers of protective exopolysaccharides such as a capsule can be present [32]. To recognize their host and pass through these layers, phages usually directly bind exposed glycans as an initial host receptor before moving down to the cell surface [10,31,33,34]. There, the virus typically binds a second so-called terminal receptor, which results in irreversible adsorption and DNA ejection [34]. The receptor-binding proteins (RBPs) that target primary and terminal receptors are typically displayed on different structures with longer, lateral tail fibers or stubby tailspikes commonly targeting primary receptors followed by central tail fibers or tail tip complexes binding terminal receptors [reviewed in reference [34].

Bacterial exopolysaccharides and the dense O-antigen layer of the LPS effectively shield the cell surface from the access of tailed phages unless the phage carries enzymatic tailspikes or glycan-targeting tail fibers to specifically bind them as part of primary receptor recognition [28,29,31,33,34] (Fig 1A). Consequently, the variability of the ca. 180 different types of O-antigen and other surface structures greatly contribute to the observed narrow host range of most enterobacterial phages [10,30,35,36]. This diversity is further increased by accessory O-antigen modification systems that can alter the polysaccharide chains, e.g., by acetylation or glycosylation [36]. However, laboratory strains of the *E. coli* K-12 lineage do not express protective exopolysaccharides such as capsules and lack even the O-antigen layer due to disruption of the *wbbL* gene by a transposon [27] (Fig 1A). Phages can therefore readily access the bacterial cell surface, which enabled broad sampling of *E. coli* phages for the original BASEL collection [10,28]. However, most of these phages are unable to infect *E. coli* K-12 with restored O16-type O-antigen with few exceptions that likely carry RBPs targeting this glycan such as *Hanrivervirus* JakobBernoulli (Bas07) or *Tequintavirus* IrisvonRoten (Bas32; Fig 1A and 1B). In addition, some phages like N4 or the *Vequintavirinae* can bypass the O-antigen layer by targeting the "N4 glycan receptor" (NGR), a recently described glycan that is broadly conserved among enterobacteria [10,17] (Fig 1A and 1B). The shielding of surface receptors by the O-antigen layer seems to be primarily due to its density because it is not more than 30 nm deep, while the tail fibers of model phages T4 (ca. 145 nm [37]), T5 (80 nm [38]), or T7 (30 nm [39]) could otherwise easily reach through it to the cell surface. However, these phages are completely blocked by restored O16-type O-antigen expression [10] (Fig 1B).

While the absence of an O-antigen barrier in the K-12 strain used for sampling the BASEL phages made it much easier to isolate many diverse phages, it also meant that phages that

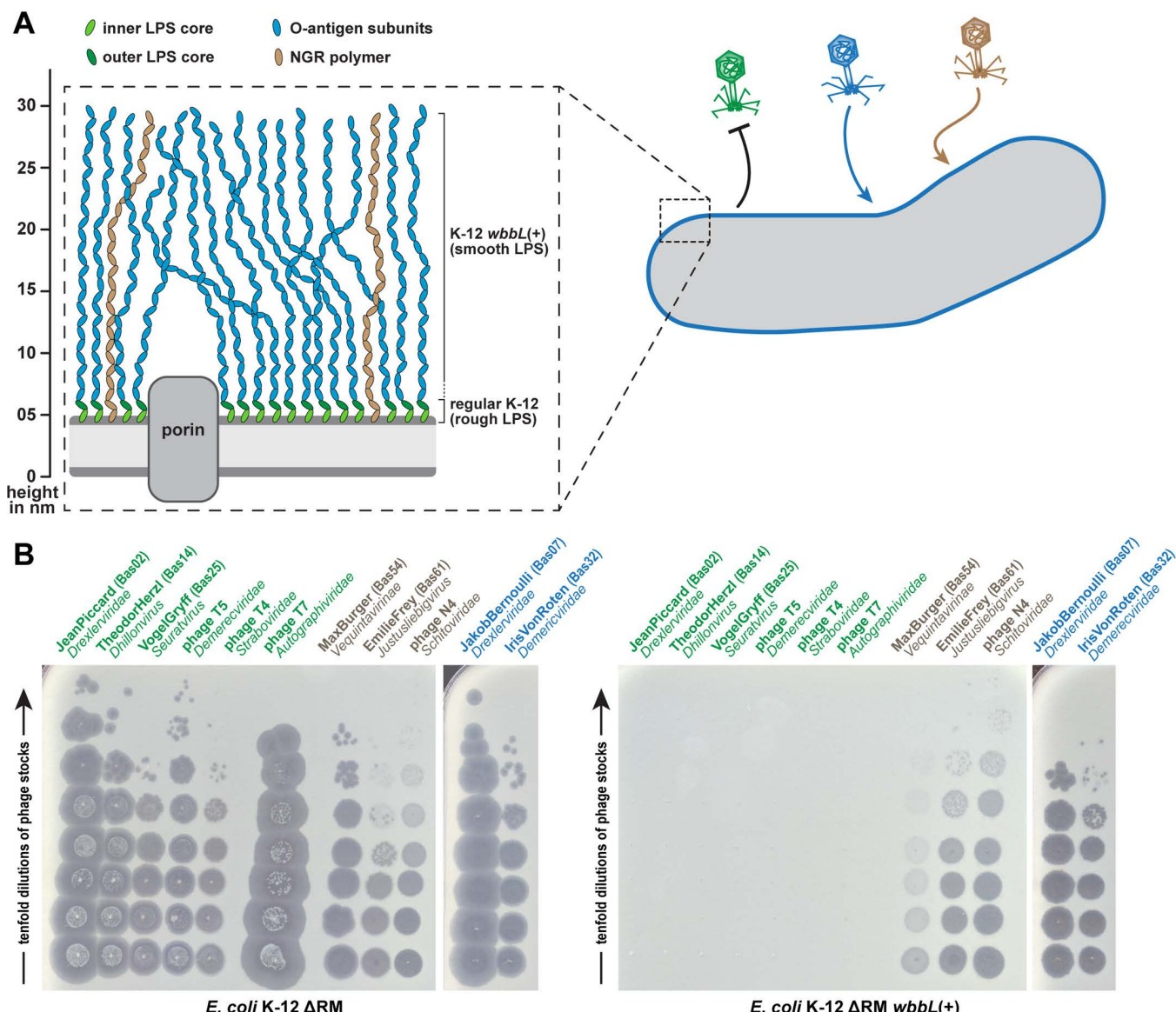

**Fig 1. O-antigen glycans as a barrier or host receptor for phage infections. (A)** The illustration shows cell surface glycans of *E. coli* K-12 laboratory strains, including rough lipopolysaccharide (LPS) without O-antigen and smooth LPS (with genetically restored O16-type O-antigen; see details in Materials and methods). The O-antigen can be a barrier to phage infection (green), used as a host receptor (blue), or be bypassed by phages that target the N4 glycan receptor (NGR) chains (brown). **(B)** Serial dilutions of different phages characterized previously [10] were spotted on the *E. coli* K-12 ΔRM strain without (left) or with (right) genetically restored O-antigen expression to illustrate the three different modes of interaction with these surface glycans (same color code as in panel A).

depend on this glycan as part of host receptor recognition could not be sampled. Using different sewage samples, we had previously shown that these are quantitatively rare, though they would of course be qualitatively interesting to expand the assorted diversity of the BASEL collection [10]. We have therefore now systematically isolated and characterized diverse phages that infect *E. coli* K-12 with O16-type O-antigen but not the isogenic strain without this glycan (Fig 2A and 2B). Fortuitously, we observed that phages depending on O16-type O-antigen as host receptor are much more common in environmental samples than in sewage (S1 Table). Most, but not all of these viruses belonged to phage groups that had already been

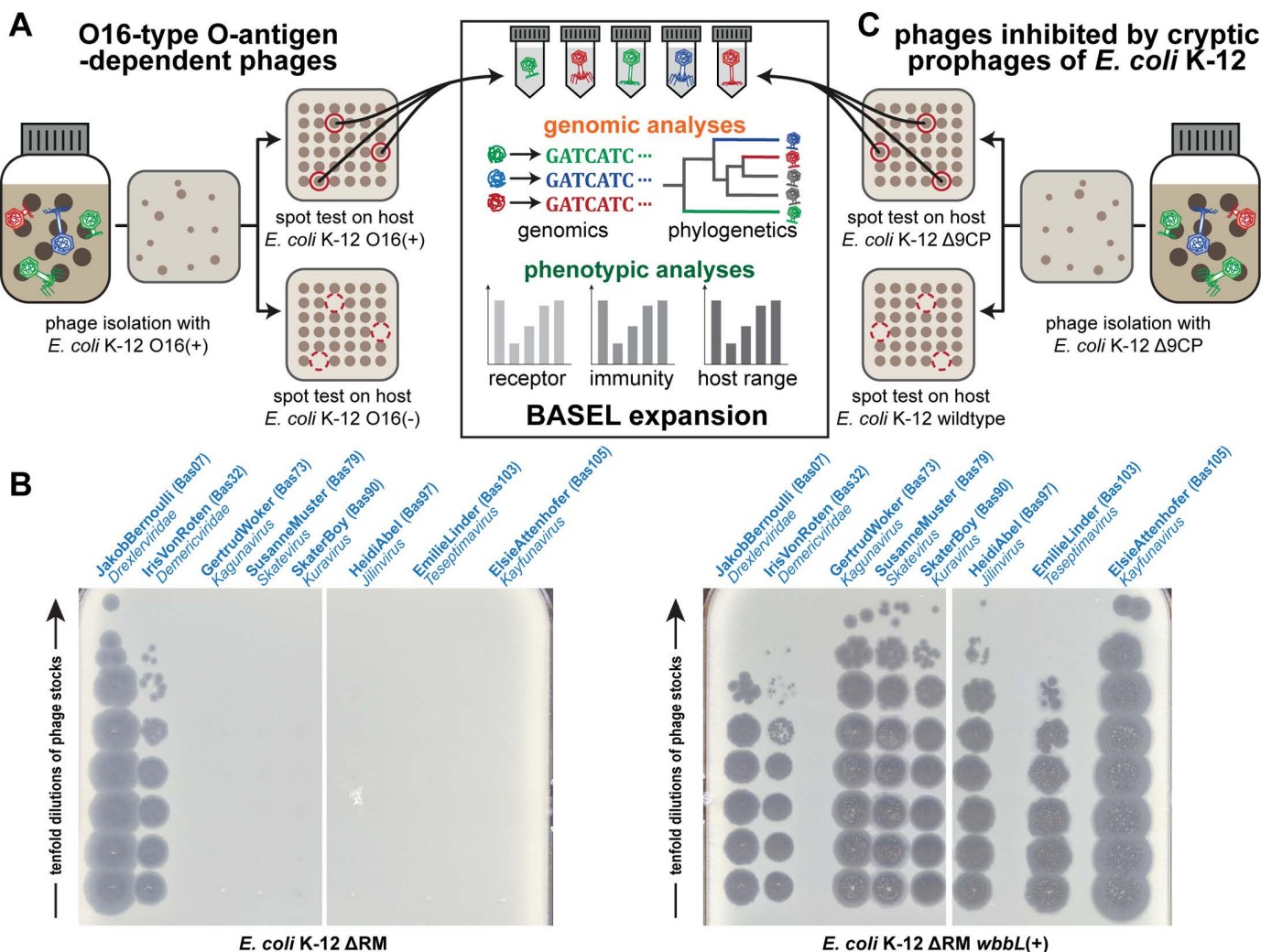

**Fig 2. Expanding the BASEL collection with phages dependent on O-antigen or inhibited by K-12 cryptic prophages.** (A) Phages dependent on O16-type O-antigen were isolated and characterized as shown in the illustration (analogous to our previous work [10]). (B) O-antigen dependence of diverse newly isolated phages characterized in this work compared to JakobBernoulli and IrisVonRoten from our previous study that can infect *E. coli* K-12 without (left) and with (right) O-antigen [10]. (C) Phages inhibited by the cryptic prophages in the K-12 genome were isolated analogously to those dependent on O-antigen (see panel A) with help of the *E. coli* K-12 Δ9CP strain (details outlined in Materials and methods).

known for their dependence on surface glycans including host O-antigen such as podoviruses of *Gamaleyavirus* genus or siphoviruses of the *Kagunavirus* genus [33] (see Fig 2B).

## Overview: Including phages inhibited by resident immunity of *E. coli* K-12 in the BASEL collection

Bacteriophages encounter multiple layers of antiviral defenses outside and inside their host cells that must be overcome for successful viral replication [40]. Some of these defenses prevent phage adsorption or DNA ejection, while others directly attack the invading virus such as restriction-modification (RM) systems, which cleave foreign DNA based on self/ non-self recognition through certain DNA modifications. Beyond direct attack, a deeper layer of defense systems protects the bacterial population through abortive infection, i.e., the altruistic suicide of infected cells prior to successful phage replication [41]. Recent work on anti-phage defense

systems has vastly expanded our knowledge about their abundance and diversity through the discovery of many new systems that target various aspects of phage biology and are countered by phages in diverse ways [40–45].

The phages of the original BASEL collection had been sampled with a derivative of the commonly used *E. coli* K-12 MG1655 laboratory strain in which we had knocked out all known RM systems, i.e., the EcoKI type I RM system and three type IV restriction systems [10]. This enabled broad sampling of *E. coli* phages, including several taxonomic groups of phages that displayed considerable sensitivity to EcoKI and would otherwise have been excluded [10]. However, the literature and commonly used prediction tools such as Defense-Finder and PADLOC indicate that *E. coli* K-12 encodes several additional *bona fide* defense systems that might limit the range of phages accessible with this genomic background [46,47] (see details at *Bacterial strains* in Materials and methods). We had not considered this to be a widely relevant factor when composing the original BASEL collection because phage defenses beyond RM systems were thought to be rather specific. However, our own results showing broad antiviral defense by the Fun/Z system and the work of others, e.g., on the Zorya system revealed that this is not necessarily the case [10,24].

Notably, all these *bona fide* defense systems are encoded on the cryptic prophages of *E. coli* K-12 that are genomic islands descended from once active prophages, i.e., the facultatively host-integrated form of so-called temperate phages [46,47]. This is in line with the common observation that anti-phage defenses are frequently encoded on mobile elements and genomic islands [48–50]. To overcome possible sampling biases caused by these *bona fide* phage-defense systems (and possibly others that are yet unknown), we thus performed phage isolation experiments with a variant of *E. coli* K-12 in which a previous study had removed all cryptic prophages, *E. coli* K-12 BW25113 Δ9CP (both with and without restored O-antigen) [51] (Fig 2C). Similar to previous work in *Vibrio* that had used an analogous approach [50], these experiments indeed resulted in the isolation of an additional diversity of phages both from groups already covered in the BASEL collection and from a genus that had never been isolated before using any *E. coli* laboratory strain (*Wifcevirus* myoviruses).

## Composition of the expanded BASEL phage collection

The original BASEL phage collection had been composed of 68 new handpicked phage isolates that we characterized alongside a panel of 10 well-studied model phages such as the classic T-phages [10]. These phages represent many major groups of *E. coli* phages, which have been previously described in phage isolation studies with *E. coli* K-12 [52,53]. The (mostly genus-level) groups comprising these phages showed characteristic and specific patterns of features regarding receptor usage, host range/ recognition, and sensitivity or resistance to bacterial immunity in our data and those from other studies [10,20–25]. These observations make it possible to learn about the properties of newly isolated phages merely from their genome sequences or taxonomic position, e.g., to inform a targeted selection or engineering of effective phages for individual cases of phage therapy [54–56].

However, a close look at published work describing phages isolated with *E. coli* hosts beyond the K-12 laboratory strain reveals important gaps in the assorted diversity of the original BASEL collection. These include, e.g., several groups of podoviruses such as the *Gordonclarkvirinae* with characteristic prolate virions [57], diverse small siphoviruses such as the genus *Kagunavirus*, and a few myoviruses such as the genus *Wifcevirus* [35,58,59]. Using *E. coli* K-12 with restored O-antigen expression and/ or without cryptic prophages, we have now isolated and characterized more than a hundred new phages that are unable to grow on the regular *E. coli* K-12 host used for the original BASEL collection. These new isolates cover all major groups of phages (and some rarely isolated ones) that have been reported to infect

*E. coli*, and we selected 37 isolates for an expansion of the BASEL collection based on their genomic and phenotypic diversity (Fig 3 and S2 Table). We systematically characterized these phages analogous to the phages in our previous study (Fig 4A–4D) and used several original BASEL phages that could overcome the O16 O-antigen barrier as controls for the phenotypic experiments. For these, no relevant differences to their phenotypes in the original paper were observed when the host strain expresses O16 O-antigen (S1B Fig). These results suggest that the O-antigen barrier and other defenses are largely orthogonal and that our results from this study are comparable to the original publication on the BASEL collection.

## Properties of siphoviral genus *Kagunavirus*

Small siphoviruses like the *Drexlerviridae* family (relatives of phage T1) or genus *Dhillonvirus* are frequently isolated using *E. coli* as a host and typically have genomes of 40–50kb size. The diverse representatives of these groups in the original BASEL collection mostly showed considerable sensitivity to RM systems and—as expected from previous work—targeted different porins as terminal host receptors [10,60,61]. Only one of these phages, JakobBernoulli (Bas07), is able to infect *E. coli* K-12 with restored O-antigen expression, suggesting that its lateral tail fibers can target O16-type O-glycans [10] (see also S1A Fig and below). Our targeted search for phages that depend on O16-type O-antigen resulted in the isolation of various small siphoviruses of genus *Kagunavirus* within the subfamily of *Guernseyvirinae* (Fig 5A–5D). These phages have small genomes of ca. 41–44kb, which makes them attractive models for synthetic biology applications, e.g., in the case of phage CM001 [62]. While all our isolates fully depend on restored O-antigen expression, only some of them (Bas70–73) infect a regular *E. coli* K-12 host while others (Bas74–76) could only be isolated using a K-12 host devoid of cryptic prophages (Fig 5D). Phenotypically, both groups of *Kagunavirus* isolates behave very similarly and display a high sensitivity to RM systems and a variable sensitivity to the Old antiphage defense system that is typical for small siphoviruses [10] (Fig 5D). The dependence on glycans as host receptor seems to be a general property of *Guernseyvirinae* phages, including, e.g., the prototypic *Kagunavirus* K1G targeting the K1 capsule of *E. coli* [15] or the *Jerseyvirus* SETP3 and several relatives targeting O-antigen of *Salmonella enterica* [63].

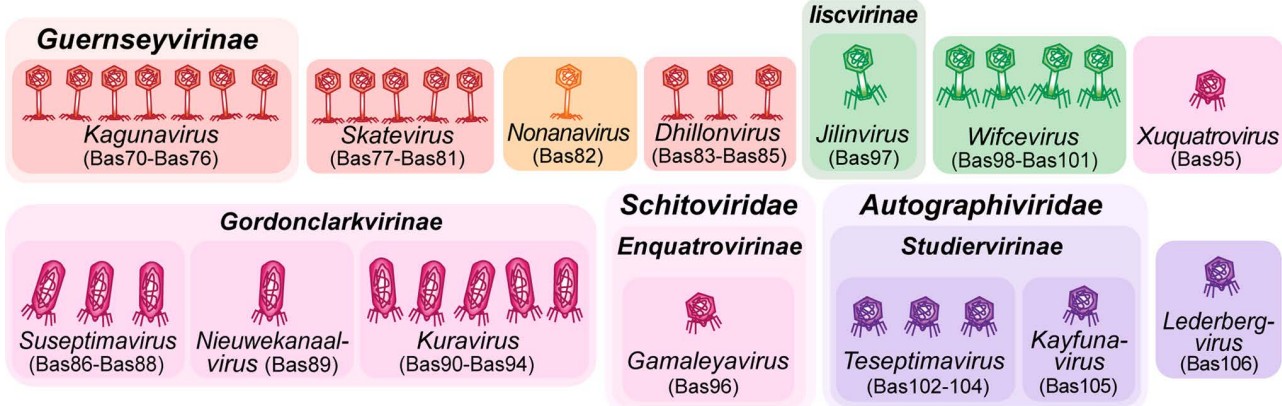

**Fig 3. Overview of new phages included in the BASEL collection.** The illustration shows new phage isolates included in the BASEL collection with their Bas## identifiers. Isolates are sorted by morphotype and their current ICTV classification [68]. The color code distinguishes phages based on morphotype and genome size. For siphoviruses, small phages (ca. 30–50 kb) are shown in red and medium-sized phages (ca. 50–100 kb) are shown in orange. Small myoviruses (ca. 30–100 kb) are shown in green. For podoviruses, small phages (ca. 30–50 kb) are shown in violet and medium-sized phages (ca. 50–100 kb) are shown in pink.

## Properties of other O16-dependent small siphoviruses and *Dhillonvirus*

Besides phages of the genus *Kagunavirus*, we also obtained different other small siphoviruses that depend on restored O16-type O-antigen to infect *E. coli* K-12 and new isolates of the *Dhillonvirus* genus (Fig 6A–6G). Some of the new O16-dependent isolates belong to a cluster of siphoviruses formed primarily by the genus *Skatevirus* (Bas77–81, Fig 6C and 6D). These viruses have genomes of 46–48 kb, which is well within the typical range for small siphoviruses. Additionally, we present phage KarinMuster (Bas82), belonging to the rarely reported *Nonanavirus* genus comprising relatives of the more commonly known *Salmonella* phage 9NA (Fig 6E) [64]. The genome size of this phage is ca. 58.5 kb and thus around 10 kb larger than that of the other small siphoviruses presented here, reminiscent of the *Queuovirinae* characterized in our previous work [10]. This latter group encodes a 7-deazaguanosine

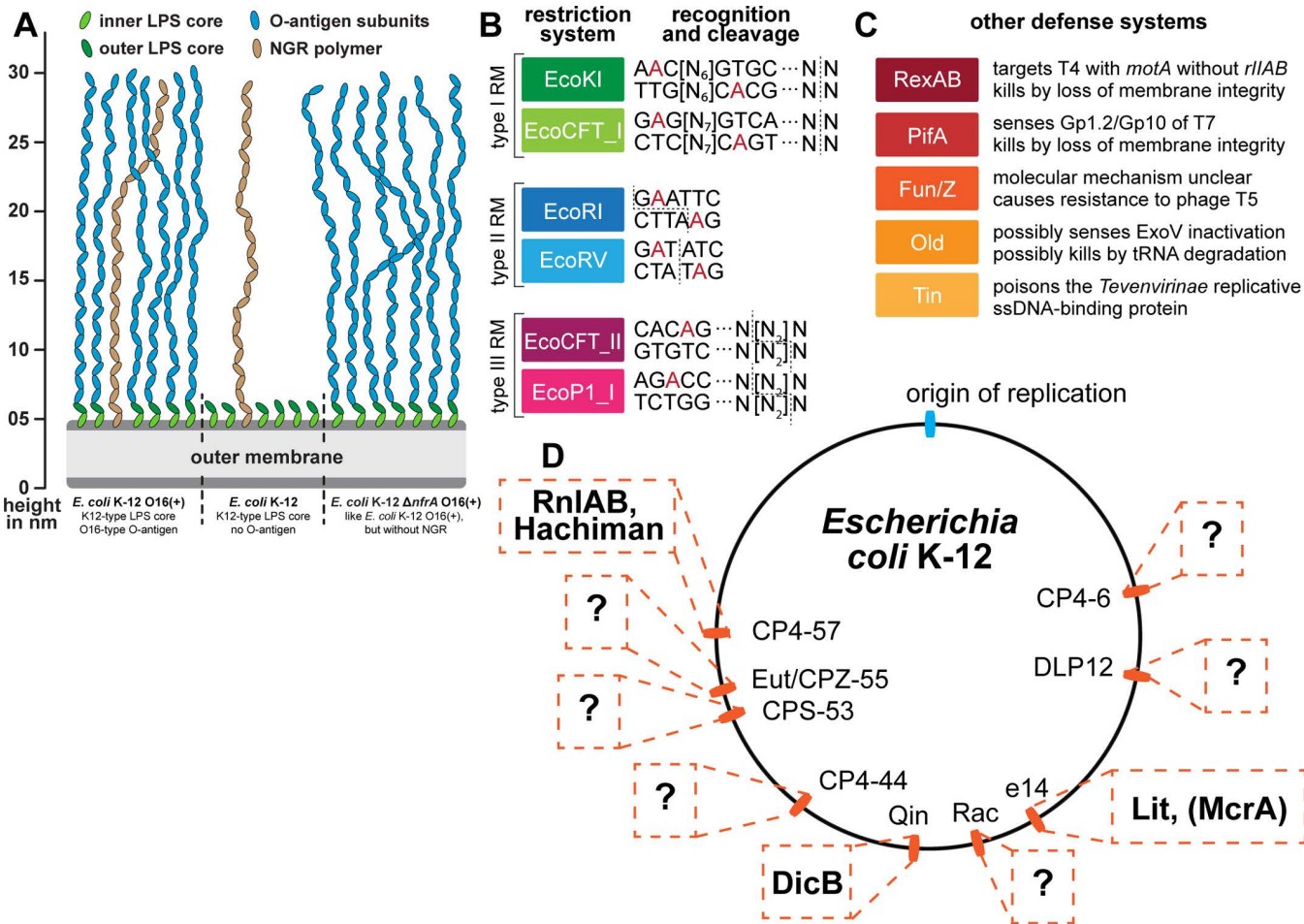

**Fig 4. Overview of *E. coli* surface glycan variants and the immunity systems used in this study.** **(A)** Surface glycans of different *E. coli* K-12 variants are shown schematically (compare Fig 1A and see further details in Materials and methods). None of the newly isolated phages in this work showed any dependence on the NGR glycan (see S1A Fig) which has thus been omitted from the other illustrations. **(B, C)** Restriction-modification (B) and other defense systems (C) used to characterize phage isolates are summarized schematically (same as used in our previous work [10]). **(D)** The genome of *E. coli* K-12 contains nine cryptic prophages that encode all remaining antiviral defense systems predicted by DefenseFinder or PADLOC after deletion of all restriction systems in our previous work (including McrA; see details on these defense systems in Materials and methods under *Bacterial strains*) [10,46,47]. Due to the known accumulation of antiviral defenses in prophages [48,49], it seems likely that also many of the possibly still unknown defense systems of *E. coli* K-12 would be encoded in these loci (indicated by question marks). The *E. coli* K-12 Δ9CP variant of Wang and colleagues [51], which we use in the current study, has been cured of all nine cryptic prophages.

modification pathway in their additional genome content which provides considerable resistance to RM systems [10,65]. However, there is no recognizable additional locus in KarinMuster and its relatives that could encode such supplemental biological functions [64], though their distant relationship to other known phages and limited knowledge about their biology makes it difficult to draw firm conclusions. Very similar to small siphoviruses characterized previously and the *Kagunavirus* phages shown above, all these phages are markedly sensitive

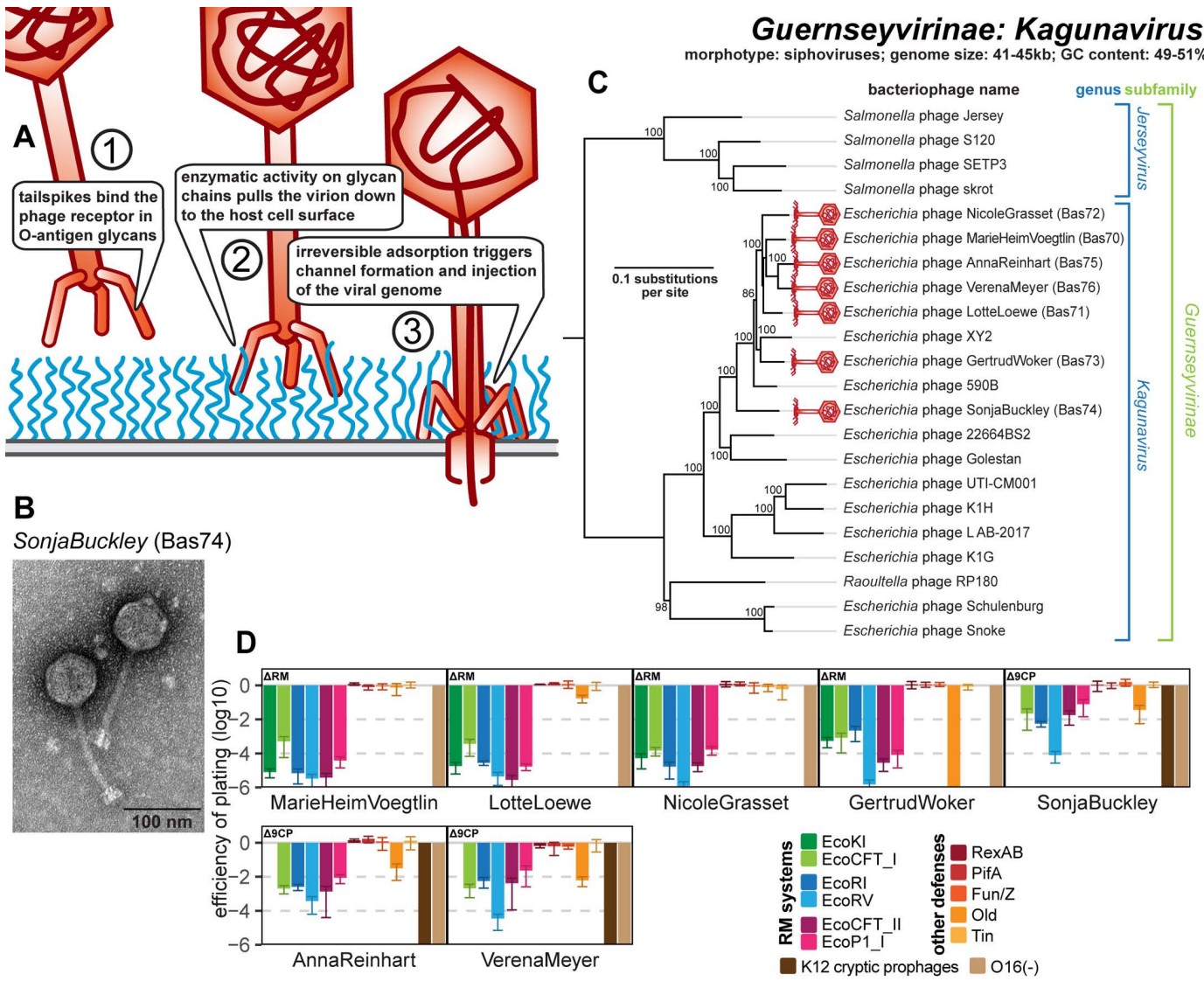

**Fig 5. Overview of *Guernseyvirinae* genus *Kagunavirus* phages.** (**A**) Schematic model of host recognition by *Kagunavirus* phages (O-antigen chains in blue). In analogy to other siphoviruses infecting *E. coli* like lambda or T5, we anticipate that these phages also use a phage-derived conduit for DNA injection across the outer membrane after the tail fibers recognizing the terminal host receptor have bent off [110,141]. (**B**) Representative TEM micrograph of phage Sonja-Buckley (Bas74). (**C**) Maximum-likelihood phylogeny of *Kagunavirus* phages and relatives based on a curated whole-genome alignment with bootstrap support of branches shown if > 70/100. Newly isolated phages of the BASEL collection are highlighted by red phage icons. The phylogeny was rooted between the genera *Kagunavirus* and *Jerseyvirus*. (**D**) The results of quantitative phenotyping experiments with *Kagunavirus* phages regarding sensitivity to altered surface glycans and bacterial immunity systems are presented as efficiency of plating (EOP). Small notes of "ΔRM" or "Δ9CP" indicate on which host strain the respective phage has been characterized (see Materials and methods). Data points and error bars represent the average and standard deviation of at least three independent experiments. Raw data and calculations are available in S1 Data.

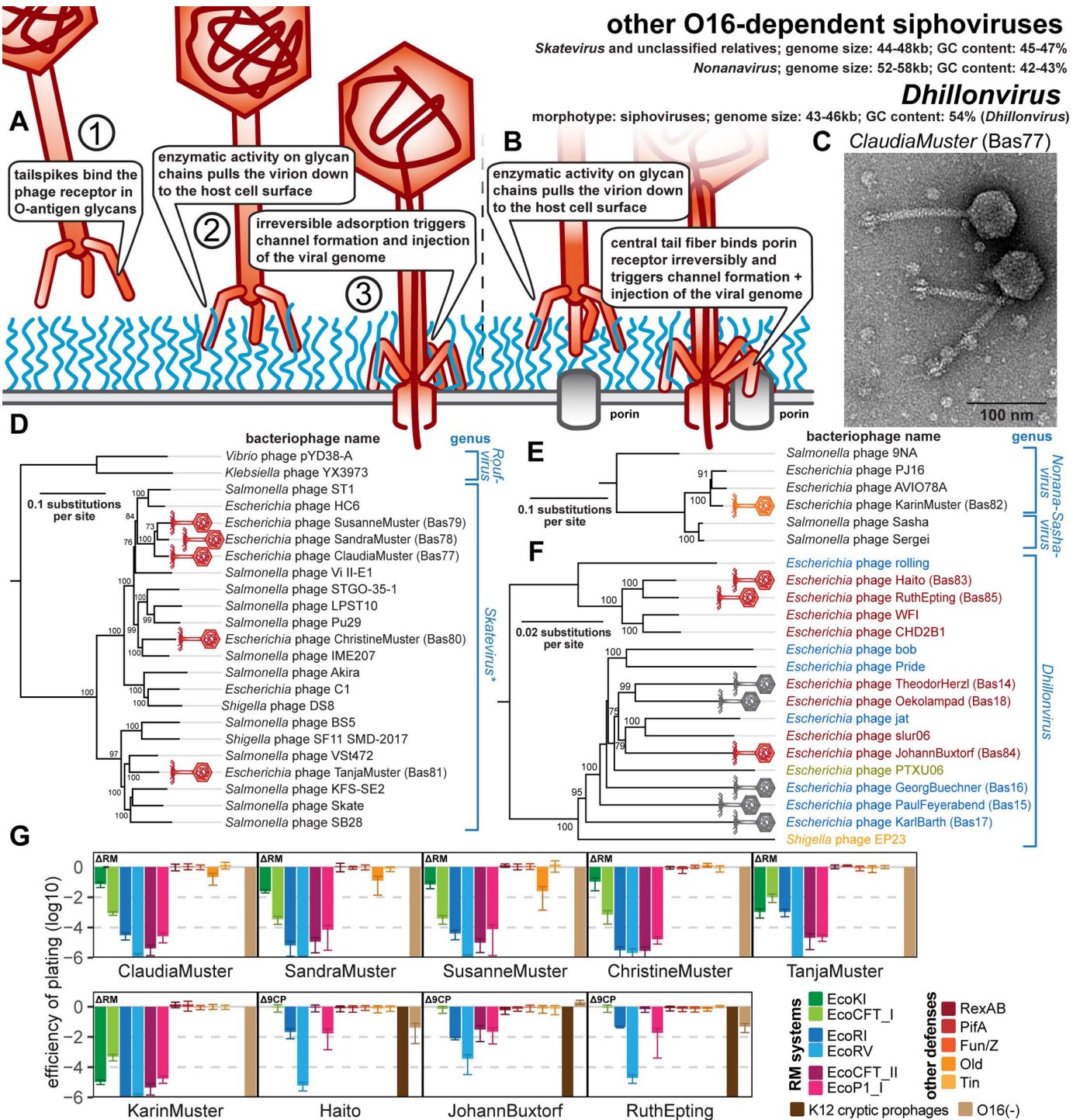

**Fig 6. Overview of other O16-dependent siphoviruses of genera *Skatevirus* and *Nonanavirus* as well as new isolates of siphoviral genus *Dhillonvirus*. (A)** Schematic model of host recognition by O16-dependent small siphoviruses just as shown for *Kagunavirus* phages in Fig 5A. **(B)** Schematic model of host recognition by small siphoviruses of the *Dhillonvirus* genus that use first a lateral tail fiber to overcome the O-antigen barrier before binding a porin receptor with the central tail fiber (instead of only glycan-targeting tailspikes for the LPS-dependent small siphoviruses; see also our previous work [10]). **(C)** Representative TEM micrograph of *Skatevirus* ClaudiaMuster (Bas77). **(D)** Maximum-likelihood phylogeny of the *Skatevirus* genus and relatives based on a curated whole-genome alignment with bootstrap support of branches shown if > 70/100. The phylogeny was rooted between genus *Roufvirus* and the others. Note that most reference phages presented here as *Skatevirus* have no formal ICTV classification while others are scattered over several seemingly paraphyletic genera, prompting us to show all phages as *Skatevirus* [68]. **(E)** Maximum-likelihood phylogeny of the *Nonanavirus* genus and relatives based on a curated whole-genome alignment with bootstrap support of branches shown if >70/100. The phylogeny was rooted between phage 9NA and the other relatives based on their much closer relationship.

Note that *Sasha* and *Sergei* are formally classified as *Sashavirus* while 9NA and AVIO78A are classified as *Nonanavirus* [68], which—according to this phylogeny—would be paraphyletic. **(F)** Maximum-likelihood phylogeny of the *Dhillonvirus* genus based on a curated whole-genome alignment with bootstrap support of branches shown if > 70/100. The phylogeny was rooted as previously [10] between the clade containing phage WFI and the others. In (D-F), newly isolated phages of the BASEL collection are highlighted by red or orange phage icons. The color code of phage names in (F) refers to porin host receptors with blue indicating FhuA, red LptD, green BtuB, and orange LamB [10] (see also S2B Fig). **(G)** The results of quantitative phenotyping experiments with *Skatevirus*, *Nonanavirus*, and new *Dhillonvirus* isolates regarding sensitivity to altered surface glycans and bacterial immunity systems are presented as efficiency of plating (EOP). Small notes of "ΔRM" or "Δ9CP" indicate on which host strain the respective phage has been characterized (see Materials and METHODS). Data points and error bars represent average and standard deviation of at least three independent experiments. Raw data and calculations are available in S1 Data.

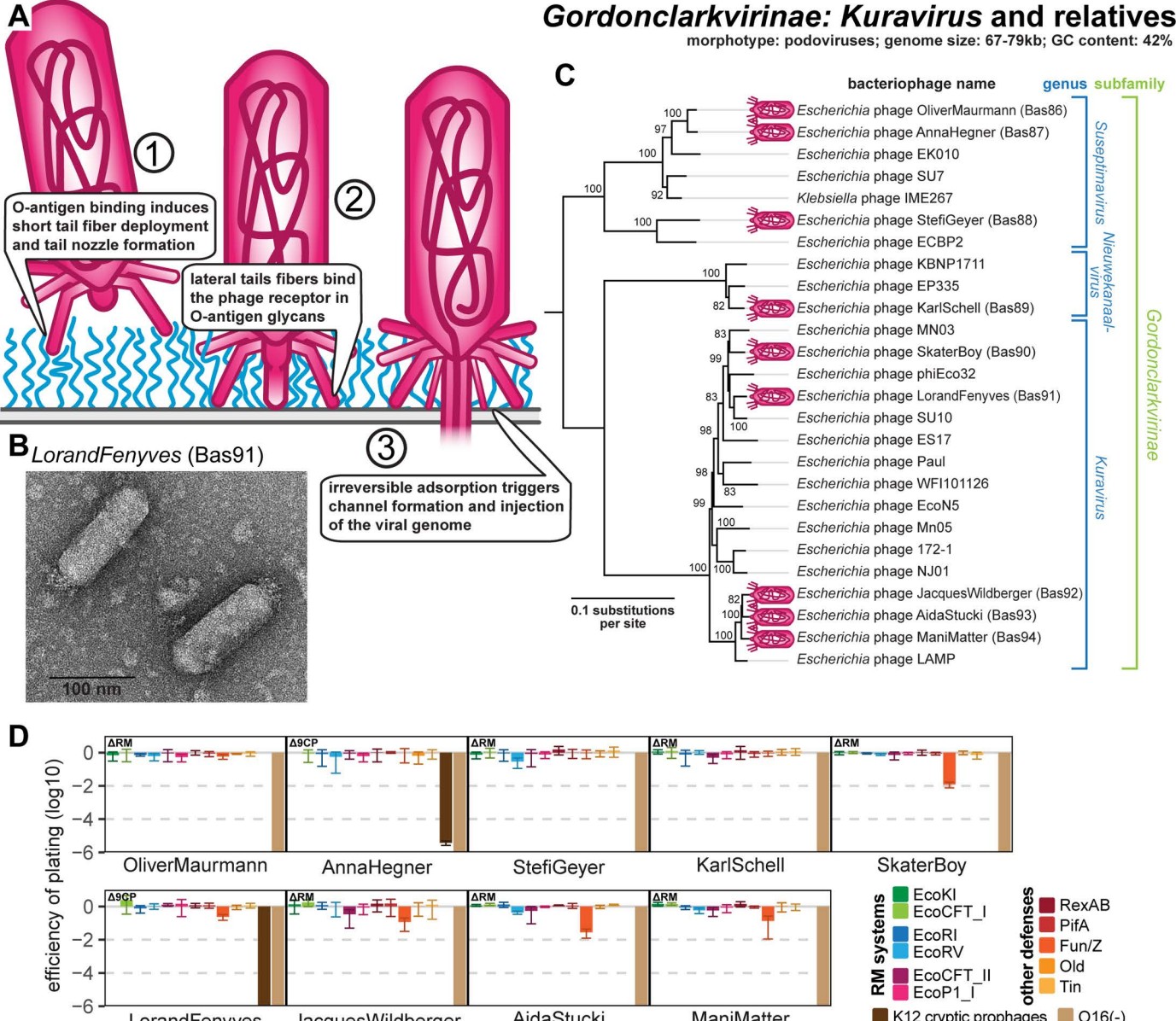

**Fig 7. Overview of *Gordonclarkvirinae* phages.** **(A)** Schematic model of host recognition by *Gordonclarkvirinae* phages as an interpretation of different molecular studies [57,117]. **(B)** Representative TEM micrograph of phage LorandFenyves (Bas91). **(C)** Maximum-likelihood phylogeny of the *Gordonclarkvirinae* subfamily based on a curated whole-genome alignment with bootstrap support of branches shown if > 70/100. Phages of the BASEL collection are highlighted by little violet phage icons. The phylogeny was rooted between the genus *Suseptimavirus* and the genera *Nieuwekanaalvirus* + *Kuravirus* based on closer relationship of the latter two. **(D)** The results of quantitative phenotyping experiments with *Gordonclarkvirinae* phages regarding sensitivity to altered

surface glycans and bacterial immunity systems are presented as efficiency of plating (EOP). Small notes of "ΔRM" or "Δ9CP" indicate on which host strain the respective phage has been characterized (see Materials and methods). Data points and error bars represent average and standard deviation of at least three independent experiments. Raw data and calculations are available in S1 Data.

to RM systems and to the Old defense system, which in the latter case results in quantifiable decrease of plaque counts only for some phages (Fig 6G). The additional genome size of KarinMuster therefore does not seem to make this phage more robust than other small siphoviruses when facing the tested examples of bacterial immunity.

Finally, the phage isolation using a host without cryptic prophages identified additional members of the *Dhillonvirus* genus (Bas83–85), which—unlike their relatives of the original BASEL collection (Bas14–18)—fail to form plaques on regular *E. coli* K-12 (Fig 6B, 6F, and 6G). These new isolates do not cluster separately within the *Dhillonvirus* genus but are rather split widely across this group, suggesting that no vertically inherited feature but rather highly variable aspects of phage biology are responsible for this difference (Fig 6F). Phenotypically, the three additional phages selected for this work behave mostly similar to the original *Dhillonvirus* phages, e.g., by showing only moderate sensitivity to tested RM systems [10] (Fig 6G). Like the other siphoviruses characterized for the original BASEL collection, these *Dhillonvirus* phages display the typical arrangement of two separate types of tail fibers with distinct receptor-binding proteins [10,34] (see below and S2A Fig). Their central tail fiber has a *bona fide* target recognition module that is predicted to encode LptD specificity (S2B Fig) [10].

## Properties of podoviruses of *Gordonclarkvirinae* genus *Kuravirus* and relatives

Besides small siphoviruses, our phage isolation with *E. coli* K-12 expressing its O16-type O-antigen resulted in multiple phages belonging to podoviruses in the *Gordonclarkvirinae* subfamily (Fig 7A–7D). These phages form characteristic prolate virions of the rare C3 morphology [66] (Fig 7B) and have been frequently isolated using diverse strains of *E. coli*. However, so far, no representatives infecting the *E. coli* K-12 or B laboratory strains have been obtained, and the molecular biology of these phages has been only poorly studied. The genome sizes of this group differ widely from 67–69kb for one clade of the *Kuravirus* genus (at the bottom of Fig 7C) to 76-77kb for other *Kuravirus* phages and genera *Suseptimavirus* as well as *Nieuwekanaalvirus* (see details for all in S2 Table). Our phylogeny largely reproduces the previously identified subgroups of *Gordonclarkvirinae*, which have in the meantime been recognized as different genera in the ICTV taxonomy [67,68] (Fig 7C). The isolates infecting *E. coli* K-12 are widely scattered over the different genera of *Gordonclarkvirinae*, suggesting that no vertically inherited factor plays major roles in the ability of these phages to infect that host (Fig 7C).

All our isolates completely depend on the restored expression of O16-type O-antigen for infection, suggesting that the O-antigen and possibly other surface glycans are the conserved host receptors of these phages (Fig 7D). This is in line with previous work reporting that spontaneous *E. coli* mutants resistant to *Kuravirus* phage ES17 had always lost O-antigen expression [69]. In one of the few dedicated studies on *Gordonclarkvirinae*, the Plevka group reported the virion structure of *Kuravirus* SU10, which—in the absence of decades-long research available for standard model phages—was critical for the functional understanding and annotation of structural genes in *Kuravirus* genomes [57]. Unlike O-antigen-dependent siphoviruses (see above) and other well-studied podoviruses such as T7 [70,71], the *Gordonclarkvirinae* have two separate *bona fide* RBPs, identifiable as long-tail fibers and short-tail fibers, that likely recognize different host receptors. These two sets of tail fibers appear to be

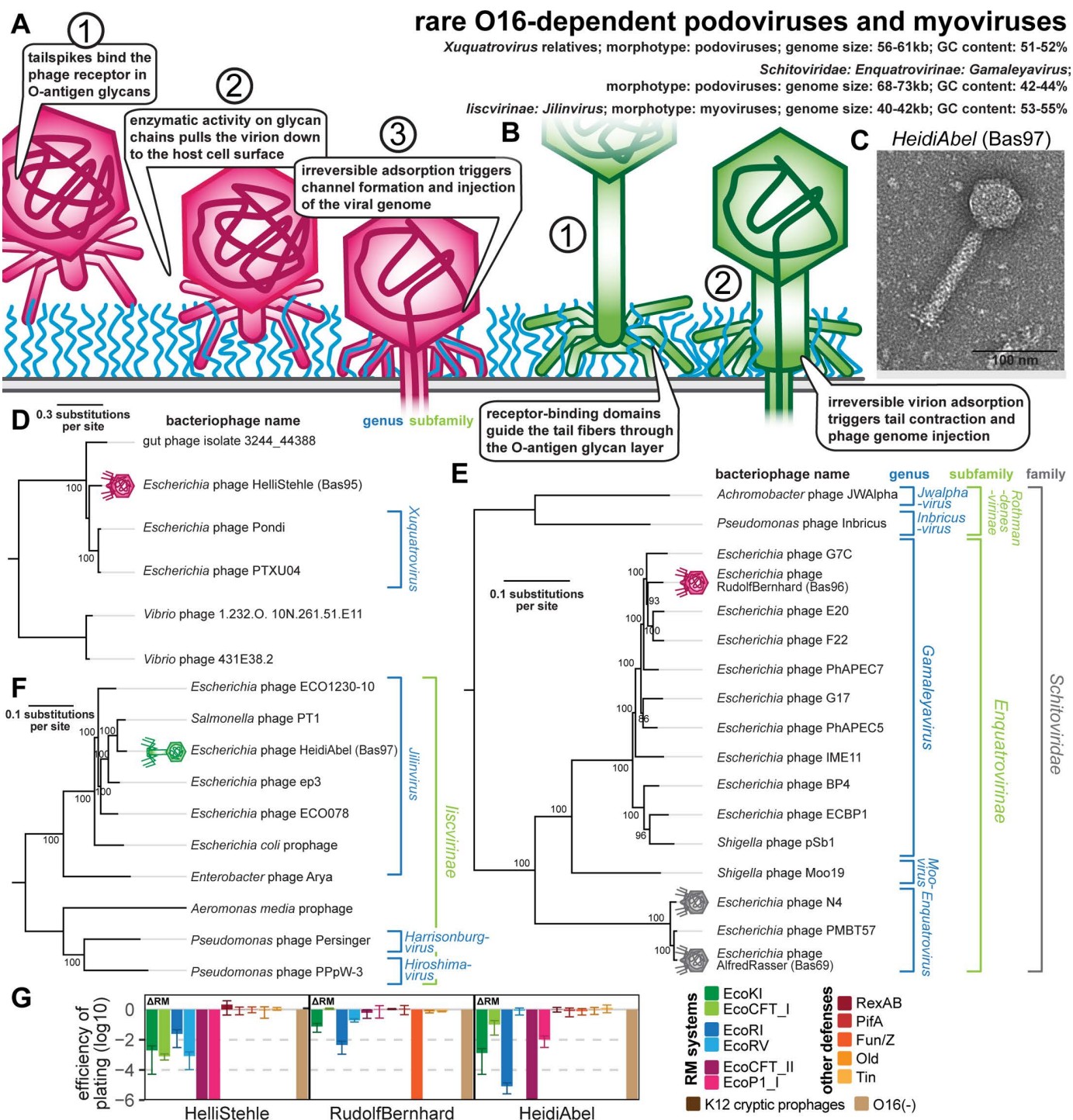

**Fig 8. Overview of rare O-antigen-dependent phage groups (*Xuquatrovirus* relatives, *Gamaleyavirus*, *Jilinvirus*).** **(A)** Schematic model of host recognition by podoviruses with enzymatic tailspikes targeting O-antigen (e.g., *Gamaleyavirus*) based on previous work on phage T7 and P22 [71,101,203]. **(B)** Schematic model of host recognition by myoviruses targeting O-antigen with a tail fiber (e.g., *Jilinvirus*). Note that *Jilinvirus* HeidiAbel encodes two sets of tail fibers (see below) whose relationship is unknown, so only a simple schematic with one set is shown. **(C)** Representative TEM micrograph of *Jilinvirus* phage HeidiAbel (Bas97). This phage forms myovirus virions with similar appearance to other Jilinvirus phages and relatives that have been studied previously [79,80,82]. **(D)** Maximum-likelihood phylogeny of the genus *Xuquatrovirus* and relatives based on a curated concatenated alignment of major capsid protein, portal protein, and terminase large subunit protein sequences with bootstrap support of branches shown if >70/100. The phylogeny was rooted between two marine *Vibrio* phages and the others based on a large phylogenetic distance between these groups. **(E)** Maximum-likelihood phylogeny of the *Enquatrovirinae* subfamily of *Schitoviridae* and relatives based on a curated whole-genome alignment with bootstrap support of branches shown if >70/100. The phylogeny was rooted between subfamilies *Rothmandenesvirinae* and subfamily *Enquatrovirinae*. **(F)** Maximum-likelihood phylogeny of the *Iiscvirinae* subfamily based on a curated whole-genome alignment with bootstrap support of branches

shown if > 70/100. The phylogeny was rooted between the genus *Jilinvirus* and the other genera. The exemplary prophages of *E. coli* and *Aeromonas media* were identified in strains USECESBL382 (NCBI GenBank AATBGZ010000031) and T0.1-19 (NCBI GenBank CP038441), respectively, using PHASTEST [204]. In (D-F), phages of the BASEL collection are highlighted by pink and green (new) or gray (old) phage icons. **(G)** The results of quantitative phenotyping experiments with the three newly described phage isolates regarding sensitivity to altered surface glycans and bacterial immunity systems are presented as efficiency of plating (EOP). Small notes of "ΔRM" or "Δ9CP" indicate on which host strain the respective phage has been characterized (see Materials and methods). Data points and error bars represent average and standard deviation of at least three independent experiments. Raw data and calculations are available in S1 Data.

used successively analogous to classical T-even myoviruses where the long-tail fibers mediate initial host recognition by targeting porins and O-antigen while the short-tail fibers subsequently bind conserved motifs in the LPS core as the terminal receptor [10,37,57,72–74] (see also below). The functionality of long and short-tail fibers of T-even phages is linked in a way that host recognition by long-tail fibers is required for the deployment of short-tail fibers and, consequently, viral genome injection [10,73,74]. Similarly, direct observation of *Kuravirus* SU10 infection using cryogenic electron tomography shows how the lateral tail fibers initially target an exposed receptor—likely O-antigen glycans—before the short-tail fibers target the cell surface and contribute to the formation of a nozzle which extends the virion tail for genome delivery [57] (see Fig 7A).

The *Gordonclarkvirinae* are remarkably resistant to RM systems despite the lack of known pathways of covalent genome modification or restriction inhibitors (Fig 7D and S2 Table). However, they encode a *bona fide* dCTP deaminase and a nucleotidyltransferase with their DNA replication genes (S3 Fig), suggesting that these phages might feature a covalent DNA modification—most simply deoxyuracil instead of thymidine like *Bacillus* phage PBS1—which may protect their genome from restriction [75]. The only detectable sensitivity to tested anti-phage defense systems was observed for Fun/Z which targets phages of the *Kuravirus* genus (Fig 7D).

## Properties of other groups of podoviruses and myoviruses

*Xuquatrovirus* relative HelliStehle (Bas95). Besides small siphoviruses and *Gordonclarkvirinae* podoviruses, all other phage groups isolated for this work were individually much rarer, and only one or a few isolates of each could be obtained. This includes two podoviruses and one myovirus that are each the only isolates of their respective groups (Fig 8A–8G). One of these is a podovirus related to the *Xuquatrovirus* genus for which there are only two closely related representatives, the original PTXU04 of the study by Korf and colleagues [58] and phage Pondi of Sidi Mabrouk and colleagues [76] (Fig 8D). These phages have an intermediate genome size for a podovirus of around 60 kb, which is ca. 20 kb larger than *Autographiviridae* (T7 relatives; see below) but ca. 15–20 kb smaller than *Gordonclarkvirinae* (see above). Our new isolate HelliStehle (Bas95) is clearly related to the *Xuquatovirus* phages with a largely colinear genome of similar size (S4 Fig) but is phylogenetically distinct (Fig 8D). The biology of these phages has largely remained enigmatic besides a genomic and proteomic analysis of PTXU04 when it had originally been described, which was crucial for a functional annotation of at least the structural genes due to low sequence similarity to well-characterized phages [58]. Consequently, a recent transcriptome-wide CRISPR interference study determined the essential genes of PTXU04 but could in the absence of additional phenotypic data not use this information for deeper insights as has been possible, e.g., with model phages T4 and T5 [77]. Our experiments show that phage HelliStehle is completely dependent on restored O-antigen expression, indicating that it is using this glycan as an essential receptor (Fig 8G). Furthermore, this phage is very sensitive to RM systems, suggesting that—unlike other larger podoviruses such as N4 or *Gordonclarkvirinae*—it lacks potent mechanisms to protect its genomic DNA (Fig 8G).

*Gamaleyavirus* **RudolfBernhard (Bas96).** Podoviruses of the *Gamaleyavirus* genus within the *Enquatrovirinae* subfamily are close relatives of the genus *Enquatrovirus*, including N4 and AlfredRasser which we have characterized previously [10]. They typically target O-antigen glycans as essential host receptors [33,78], and our isolate RudolfBernhard (Bas96) is also completely dependent on restored O16 O-antigen expression to infect *E. coli* K-12 (Fig 8G). This isolate is closely related to previously studied phage G7C (targeting O22-type O-antigen; Fig 8E) but encodes different tailspikes that likely mediate O-antigen recognition (see in the dedicated section below) [78]. Consequently, RudolfBernhard does not depend on the NGR glycan unlike N4 and AlfredRasser which target this enigmatic polysaccharide as their essential host receptor (Figs 8G and S1B) [10,17]. Like its relatives N4 and AlfredRasser, RudolfBernhard shows considerable resistance to RM systems despite not encoding any known genome protection mechanism, but unlike those phages it does have a detectable sensitivity to some systems (Fig 8G and S2 Table). It is also sensitive to the Fun/Z defense system, which targets most phages with a genome length above ca. 60 kb [10]), but the molecular basis of this antiviral defense system has remained unknown.

*Jilinvirus* **HeidiAbel (Bas97).** The genus *Jilinvirus* of the *Iiscvirinae* subfamily of myoviruses comprises only a few isolates that have not been well characterized but that all infect enterobacteria (Fig 8B and 8F). Related genera infect other hosts such as pseudomonads (Fig 8F). These phages have very small genomes for phages of the myovirus morphotype (41–44 kb) and their lifestyle has remained enigmatic. On one hand, previous work has never observed any temperate behavior of *Jilinvirus* phages or their relatives [79–82], and also our isolate HeidiAbel (Bas97) forms regular clear plaques with no signs of lysogen growth (S5A Fig). On the other hand, some *Jilinvirus* phages including HeidiAbel feature seemingly intact genes encoding an integrase and a cI-like repressor—indicative of a temperate lifestyle - which are deteriorated or absent in other representatives as noted previously (S5B and S5C Fig) [81,82]. In addition, diverse bacterial genomes contain *bona fide* prophages that would be classified in this viral group (see examples in Fig 8F) [79]. It thus appears that these phages may have a facultatively temperate lifestyle that is lost in different representatives, possibly as an adaptation to mostly lytic growth. We therefore also interpret HeidiAbel as a virulent phage, though it would be interesting to see future work exploring if or when the phage makes use of its seemingly intact lysogeny control module. Notably, there also appear to be temperate members of the well-studied *Autographiviridae* family of virulent podoviruses which may even be their ancestral state [83], suggesting that such a transition between temperate and virulent lifestyles would not be unprecedented.

Our data show that HeidiAbel is very sensitive to RM systems except EcoRV for which its genome does not contain a recognition site (Fig 8G and S2 Table). This sensitivity is shared with all temperate phages that we have tested in our previous work, but it is also very common among virulent phages with small genomes that encode no dedicated protection mechanisms (see, e.g., the small siphoviruses presented in this study or our previous work [10]). The infectivity of HeidiAbel depends totally on the presence of restored O16-type O-antigen on *E. coli* K-12, suggesting that its tail fibers target this glycan as an essential host receptor (Fig 8G).

## Properties of myoviruses in genus *Wifcevirus*

Phages of the genus *Wifcevirus* within *Caudoviricetes* have been described comparably rarely with currently less than 20 sequenced isolates in NCBI GenBank that all infect *E. coli* or *Shigella* (Fig 9A–9E). They have genomes of ca. 66–72 kb and were described as a genus when two new isolates, WFC and WFH, had been linked to several phages that had been sequenced earlier [58]. Interestingly, distant relatives of these phages in the *Pbunavirus* genus are among the *P. aeruginosa* phages most used for personalized therapy [84,85]. Based on clear homology,

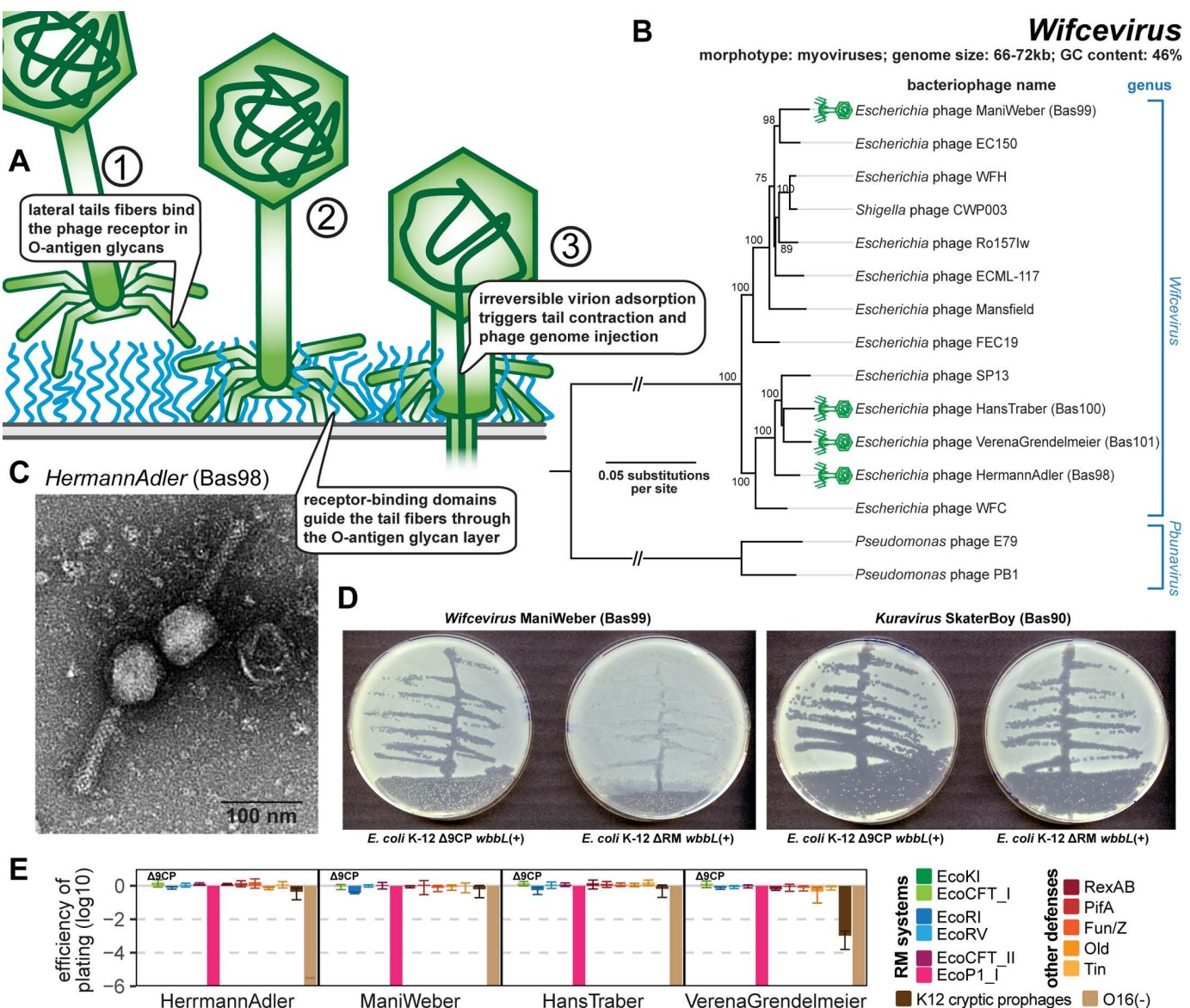

**Fig 9. Overview of *Wifcevirus* phages.** **(A)** Schematic model of host recognition by *Wifcevirus* myoviruses targeting host O-antigen (blue) using tail fibers. Note that these phages carry two sets of tail fibers (see below) whose relationship is unclear, so only a simple schematic with one set is shown. **(B)** Maximum-likelihood phylogeny of the *Wifcevirus* genus and relatives based on a curated whole-genome alignment with bootstrap support of branches shown if > 70/100. The phylogeny was midpoint-rooted between the distantly related *Wifcevirus* and *Pbunavirus* genera. Phages of the BASEL collection are highlighted by little green phage icons. **(C)** Representative TEM micrograph of phage HermannAdler (Bas98) with a virion morphology very similar to other representatives that have been visualized previously [58]. **(D)** *Wifcevirus* isolate ManiWeber (Bas99) was streaked for single plaques on *E. coli* K-12 strains with and without cryptic prophages. *Kuravirus* SkaterBoy (Bas90; see Fig 7) is shown as a control without sensitivity to the presence of cryptic prophages. **(E)** The results of quantitative phenotyping experiments with *Wifcevirus* phages regarding sensitivity to altered surface glycans and bacterial immunity systems are presented as efficiency of plating (EOP). Small notes of "ΔRM" or "Δ9CP" indicate on which host strain the respective phage has been characterized (see Materials and methods). Data points and error bars represent average and standard deviation of at least three independent experiments. Raw data and calculations are available in S1 Data.

the cryogenic electron microscopy structure of virions of a *Pbunavirus* has greatly helped us to annotate the structural genes of *Wifcevirus* phages [86]. No *Wifcevirus* infecting *E. coli* laboratory models K-12 or B has ever been reported, limiting our knowledge about the biology of these phages and their interaction with the host. However, one previous study on two

*Wifcevirus* isolates reported remarkably fast replication with only 3–5 min between adsorption and release of progeny [87]. This is much faster than even mutants of phage T7—another phage considered to be fast—which had been evolved for rapid replication but did not get faster than seven minutes in comparable assays [88]. This property of *Wifcevirus* phages could be of great interest for biotechnology or clinical applications, but the underlying molecular mechanisms have not been explored.

Our four isolates (Bas98–101) belong to both major subgroups of *Wifcevirus* phages (Fig 9B) and all depend totally on restored O16-type O-antigen as host receptor (Fig 9E), suggesting that their tail fibers target this glycan as host receptor (see also the dedicated section below). This is reminiscent of their *Pbunavirus* relatives which are known to target the LPS of *P. aeruginosa* as host receptor [86,89,90]. All our isolates could only be obtained using *E. coli* K-12 without cryptic prophages as host and their plaque formation is severely impaired in their presence (Fig 9D), though a quantitative reduction of plaque counts was only observed for VerenaGrendelmeier (Bas101; Fig 9E). All isolates are not sensitive to any tested bacterial defense system including RM systems despite the lack of any known genome protection mechanism (Fig 9E and S2 Table). One notable exception is the EcoP1_I type III RM system, to which they are very sensitive even though they have similar numbers of recognition sites for this system as for the EcoCFT_II type III RM system to which they are highly resistant (Fig 9E and S2 Table).

## Properties of *Autographiviridae*: *Studiervirinae* genera *Teseptimavirus* and *Kayfunavirus*

Podoviruses of *Autographiviridae* subfamily *Studiervirinae* are well-known from classical representatives T3 (*Teetrevirus* genus) and T7 (*Teseptimavirus* genus) and have been frequently isolated against diverse enterobacterial hosts. In our previous work, we characterized T3 and T7 together with several new isolates from different genera of this subfamily [10]. As also suggested by the literature, we found that all these phages depended on the LPS core of *E. coli* K-12 as a host receptor when infecting this host, though they were invariably unable to infect this host when O-antigen expression was restored [10,91]. These phages exhibited considerable flexibility in their glycan recognition in a way that truncations of the LPS core reduced infectivity but that usually no individual truncation fully abolished it [10,91]. Given the ubiquity of O-antigen expression among wild *E. coli* strains and its barrier function, it seems likely that these phages have evolved to target these or other surface glycans as host receptor and more accidentally recognize also the LPS core of *E. coli* K-12. Previous work has already well established that diverse representatives of *Autographiviridae* target specific types of O-antigen as host receptor which determines the host range of these phages [29,33,92–94].

In the current study, we indeed isolated several different *Teseptimavirus* phages that depend on restored O16-type O-antigen expression of *E. coli* K-12 and included three of these in the expanded BASEL collection (Bas102–104; Fig 10). Furthermore, we isolated ElsieAttenhofer (Bas105), an O16-dependent phage of the distantly related *Kayfunavirus* genus that also contains various representatives known to target host glycans such as phage K1F targeting the K1 capsule [95] or CLB_P1 targeting the O104-type O-antigen of *E. coli* strain 55989 [96] (Fig 10B). Our four new isolates encode a single tailspike or tail fiber in their genomes analogous to other *Autographiviridae* phages and likely use it to target O16-type O-antigen (see in the dedicated section below). Like their relatives, they have genomes of ca. 40kb size and are sensitive particularly to type II and type III RM systems if they have a considerable number of recognition sites (Fig 10E and S2 Table) [10]. Beyond restriction, studies on model phages T3 and T7 as well as our previous work suggested that *Autographiviridae* phages are specifically targeted by the PifA defense protein dependent on the allelic variant of the *gp1.2* dGTPase

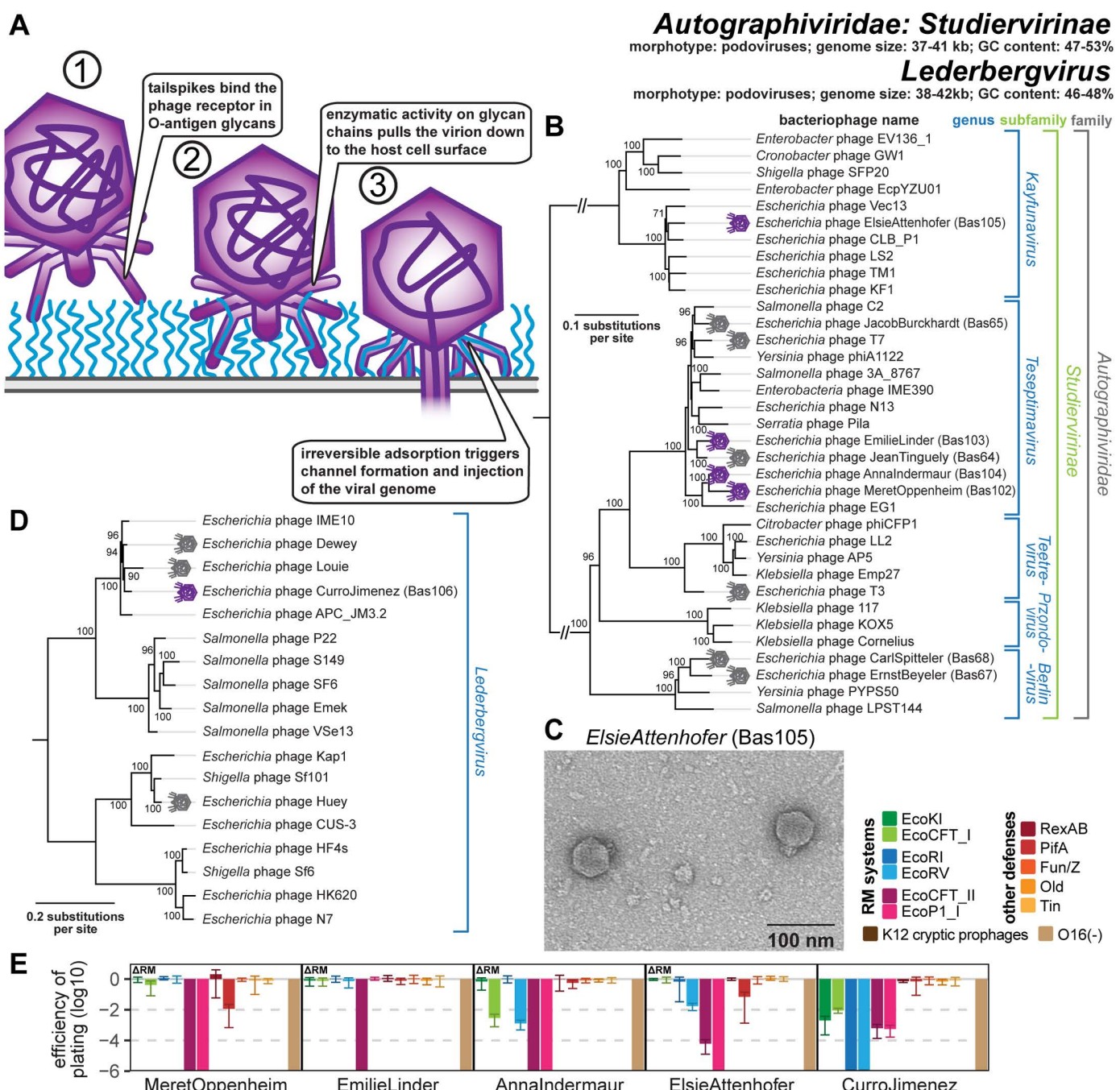

**Fig 10. Overview of *Autographiviridae: Studiervirinae* and *Lederbergvirus* phages.** **(A)** Schematic model of host recognition by podoviruses with enzymatic tailspikes targeting O-antigen (e.g., *Kayfunavirus* and *Lederbergvirus*) based on previous work on phage T7 and P22 [71,101,203]. **(B)** Maximum-likelihood phylogeny of the *Studiervirinae* subfamily of *Autographiviridae* based on a curated whole-genome alignment with bootstrap support of branches shown if > 70/100. The phylogeny was midpoint-rooted between distantly related genus *Kayfunavirus* and the other genera. Phages of the BASEL collection are highlighted by violet (new) or gray (old) phage icons. **(C)** Representative TEM micrograph of *Kayfunavirus* ElsieAttenhofer (Bas105). **(D)** Maximum-Likelihood phylogeny of the *Lederbergvirus* genus based on an alignment of their structural genes with bootstrap support of branches shown if >70/100. This phylogeny was midpoint-rooted between two major clades of this genus. Phage CurroJimenez (Bas106) is highlighted by a violet phage icon while phages Huey, Dewey, and Louie are highlighted by gray phage icons. **(E)** The results of quantitative phenotyping experiments with *Autographiviridae* phages regarding sensitivity to altered surface glycans and bacterial immunity systems are presented as efficiency of plating (EOP). Small notes of "ΔRM" or "Δ9CP" indicate on which host strain the respective phage has been characterized (see Materials and methods). Data points and error bars represent average and standard deviation of at least three independent experiments. Raw data and calculations are available in S1 Data.

inhibitor gene [10,97,98]. Of the three new *Teseptimavirus* phages, only MeretOppenheim (Bas102) encodes a *gp1.2* allele that is similar to those of sensitive phages T7 and JacobBurckhardt (Bas65), and it is indeed significantly inhibited by PifA (Figs 10E and S6A). However, also the *Kayfunavirus* ElsieAttenhofer (Bas105) is sensitive to PifA (Fig 10E), but it carries a distinct variant of the *gp1.2* gene (S6A Fig). These results suggest that *Autographiviridae* sensitivity to PifA is more complex than two allelic variants of one locus and/ or that additional factors such as the major capsid protein may play a role (as indicated by previous work [98]).

## Properties of podoviruses in genus *Lederbergvirus*

Phages in the genus *Lederbergvirus* are among the most well-known phages that use tailspikes to target enterobacterial surface glycans and particularly the O-antigen as host receptors (Fig 10A) [33,99,100]. These phages are podoviruses with genomes of around 40 kb size and a temperate lifestyle typically governed by lambda-like lysogeny control elements comprising a cI-like repressor, cII, and Cro [99]. The prototypic *Lederbergvirus* is phage P22 that has become a key model for viral host interaction. This phage targets the O:4 O-antigen of its *Salmonella* Typhimurium host with its tailspike before a tail needle penetrates the outer membrane and eventually a *trans*-envelope channel for DNA injection is formed (Fig 10A) [101,102]. Similarly, well-known *Lederbergvirus* phages like *E. coli* phage HK620 and *Shigella* phage Sf6 bind specific O-antigen variants of their respective hosts, while others target different glycans like the K1 capsule in the case of *E. coli* phage CUS-3 [103,104]. Its potency in the transduction of genes between *Salmonella* Typhimurium strains has made P22 a crucial tool for research on this organism [105], and the propensity of this phage for lateral transduction due to the initiation of viral replication and packaging before excision from the host genome can result in extremely high rates of gene transfer [106].

Beyond these classical models, *Lederbergvirus* phages infecting enterobacterial hosts are known to be common and diverse [107]. It was thus unsurprising that we isolated three different phages of this genus—Huey, Dewey, and Louie—which showed homogeneous and highly turbid plaques indicative of lysogen formation and self-immunity (S6B Fig; NCBI GenBank accessions PQ850600, PQ850609, and PQ850616) [108]. However, we also isolated one representative called CurroJimenez (Bas106) that did not show temperate behavior when infecting *E. coli* K-12 with restored O-antigen and was included in the BASEL collection that focuses on virulent phages (S6B Fig). Since it does not have obvious defects in its lysogeny control genes, we speculate that *E. coli* K-12 may lack the specific *attB* site which would be targeted by the integrase of this phage. The phylogeny of *Lederbergvirus* phages shows different distinct clades among which CurroJimenez, Dewey, and Louie are closely related in a sister group of P22 itself, while Huey is found in a more distant clade closer to CUS-3 and also HK620 (Fig 10D). However, the high genomic mosaicism of this viral genus makes it difficult to draw conclusions about the relationships of individual phages beyond the specific genes that have been used to calculate the phylogeny [33,109]. Phage CurroJimenez totally depends on the expression of restored O16-type O-antigen for the infection of *E. coli* K-12, which identifies this glycan as the host receptor targeted by its tailspike (Fig 10E). The phage is also highly sensitive to restriction-modification systems, which is not unexpected given the absence of any predictable anti-restriction mechanisms (Fig 10E and S2 Table) and seems to be common among temperate phages [10].

## Tailspikes and tail fibers to overcome the O16-type O-antigen barrier

Since the O-antigen barrier is ubiquitous among enterobacteria, it must be overcome or bypassed by any phage infecting these hosts (Fig 1). For classical model phages, the O-antigen

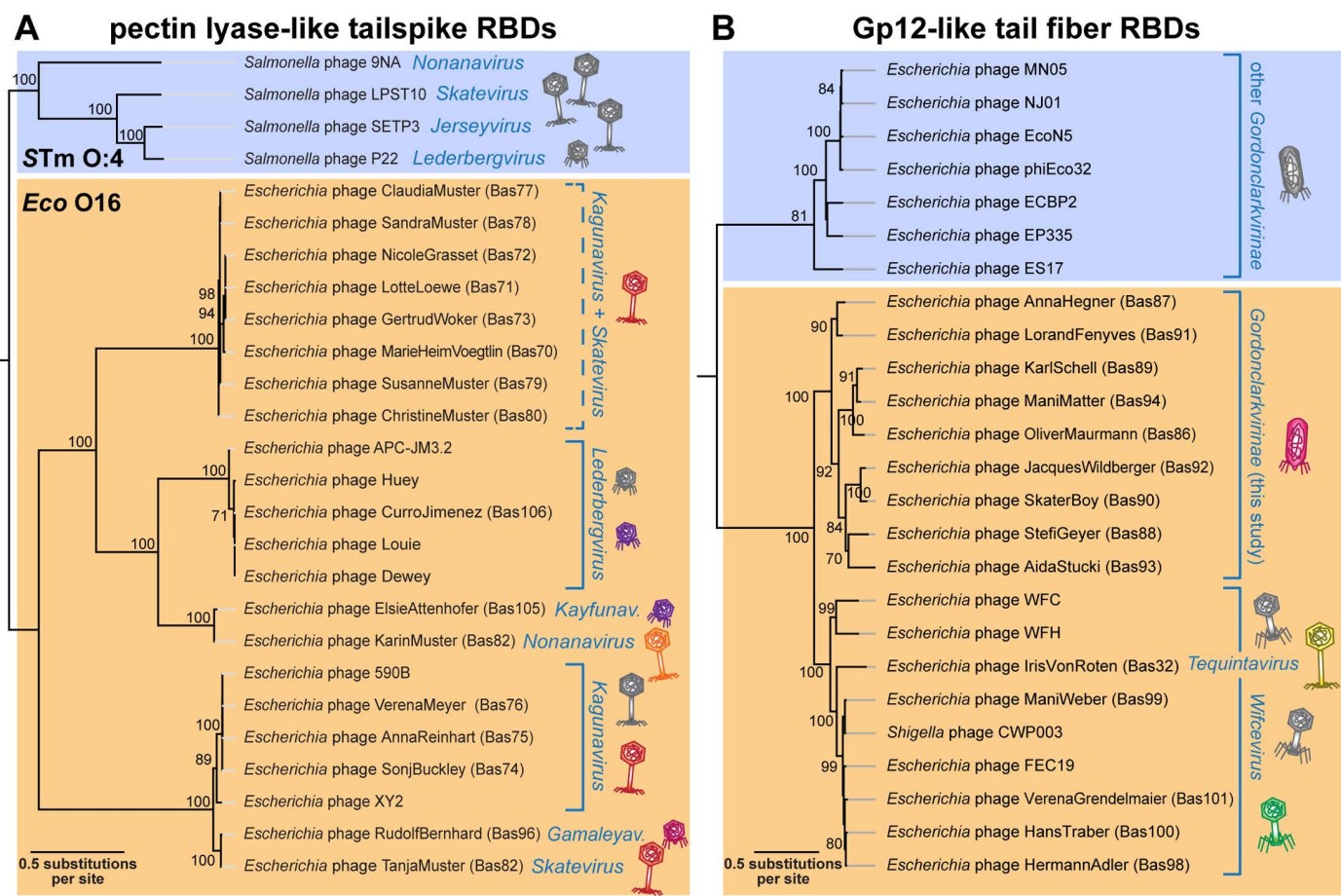

**Fig 11. Phylogenetic analyses of tailspikes and tail fibers. (A)** Maximum-likelihood phylogeny of pectin lyase-like receptor-binding domains (InterPro accession IPR011050 [114]) of tailspikes encoded by our new BASEL phages and relevant controls. The phylogeny was midpoint-rooted between previously characterized tailspikes targeting the O:4 O-antigen of *Salmonella* Typhimurium (blue) and those of our new isolates targeting *E. coli* O16-type O-antigen (orange; including a few additional previously published phages with very similar tailspikes). Bootstrap support of branches is shown if >70/100. Phages of the BASEL collection are highlighted by colorful phage icons. **(B)** Maximum-likelihood phylogeny of Gp12-like receptor-binding domains (InterPro accession SSF88874 [114]) of tail fibers encoded by our new BASEL phages and relevant controls. The phylogeny was midpoint-rooted between a clade of distant *Gordonclarkvirinae* tail fibers at the top (blue) and those of *Wifcevirus*, *Tequintavirus*, and the *Gordonclarkvirinae* presented in this study (orange). Note that a clade with *Wifcevirus* receptor-binding domains (bottom) includes diverse phages from the database as well as *Tequintavirus* IrisVonRoten (Bas32 [10]) scattered between the *Wifcevirus* isolates characterized in the current study.

has been seen as a facultative first receptor for initial host recognition followed by the binding to a second, terminal receptor for irreversible adsorption with a distinct receptor-binding protein [29,34]. As an example, phage T5 efficiently infects the *E. coli* K-12 model strain—not expressing any O-antigen—by directly targeting the FhuA porin as a terminal receptor with its central tail fiber [110]. While its lateral tail fibers are not required to infect a host without O-antigen, increased expression of the polymannose O8-type O-antigen that they target results in concomitantly faster adsorption of T5 to these hosts [111]. Conversely, increased expression of O8-type O-antigen results in decreasing adsorption of a T5 mutant specifically lacking these lateral tail fibers [111]. Restoration of the O16-type O-antigen of *E. coli* K-12 (which is not bound by the lateral tail fibers of T5) results in complete abrogation of T5 infections (Fig 1B and our previous work [10]). Similar results have been obtained for other phages of different morphotypes that express two distinct RBPs for successive recognition of O-antigen and a terminal cell surface receptor ([recently reviewed in reference 29]), and the

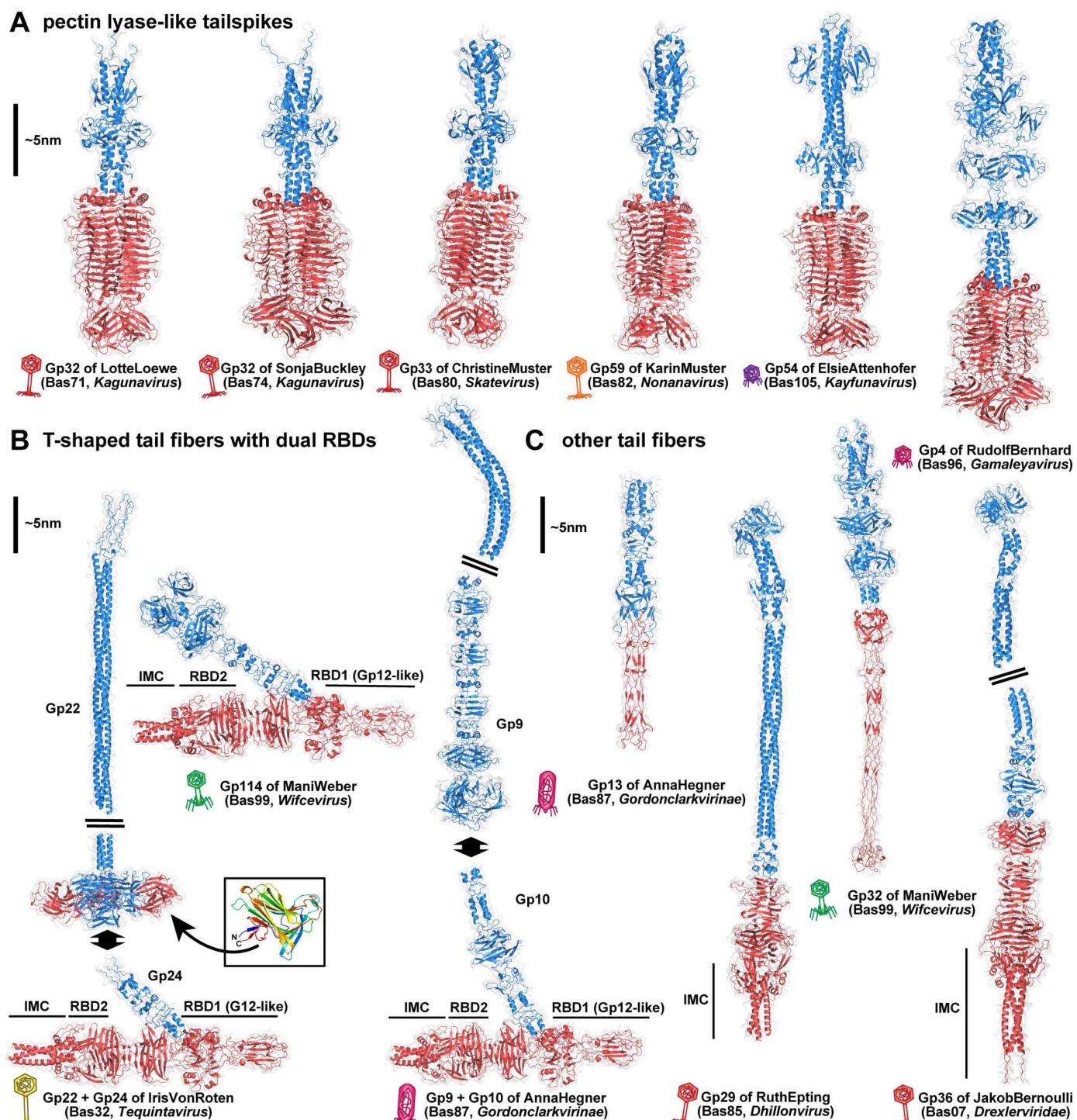

**Fig 12. Predicted structures of representative tailspikes and tail fibers.** (**A**) AlphaFold-Multimer [198] predicted structures of homotrimeric tailspikes from selected BASEL phages (see also Fig 11A). (**B**) Predicted structures of the representative T-shaped tail fiber complexes of IrisVonRoten (Bas32, *Tequintavirus*), AnnaHegner (Bas87, *Gordonclarkvirinae*), and ManiWeber (Bas99, *Wifcevirus*) which all feature the dual Gp12-like RBD1 and T5-like RBD2 (predicted with the C-terminal intramolecular chaperone domain (IMC) attached; see also Figs 11B and S10). While ManiWeber features an N-terminal anchor domain for baseplate attachment, the T-shaped tail fibers of IrisVonRoten and AnnaHegner attach to proximal fibers (Gp22 and Gp9, respectively), which connect the complete fiber structure to the distal baseplate complex. The proximal fiber protein Gp22 of IrisVonRoten contains an additional galectin-like domain (inset box; colored N- (blue) to C- (red) terminus) that protrudes from the C-terminus of the fiber and could contribute to host receptor interaction as an additional RBD. While the overall setup of the T-shaped tail fibers is highly conserved, the RBDs are highly variable and seem to be frequently transferred horizontally (Figs 11B, S9C, and S10D). (**C**) Predicted structures of other tail fibers with single RBDs (more shown in S7 Fig). The short-tail fiber (Gp13) of AnnaHegner (Bas87,

*Gordonclarkvirinae*) and the second tail fiber (Gp32) of ManiWeber (Bas99, *Wifcevirus*) have an elongated tip similar to the long-tail fiber of phage T4 including conserved HxH motifs for ion coordination [118]. Tail fibers of JakobBernoulli (Bas07, *Drexlerviridae*; Gp36) and RuthEpting (Bas85, *Dhillonvirus*; Gp29) both contain elongated N-terminal coiled coils connected to receptor-binding T5-like fiber tips with C-terminal IMC domains. In (A)-(C), N-terminal anchor domains and fiber shafts are colored blue, while the RBDs are colored red. A ~5-nm vertical scale bar is provided for reference. In all panels, viral morphotypes are annotated by phage icons analogous to Fig 11A and 11B.

concept also applies to analogous recognition of other initial receptors like capsules [112]. However, previous work had already indicated that some phages—primarily podoviruses like classical models T7 and N4—can also merely depend on a single receptor such as O-antigen or other glycans for host recognition and irreversible adsorption [17,29,31,70]. Similar to these, the O16-dependent phages characterized in this study typically feature a single RBP to target this glycan, seemingly for both initial host recognition and subsequent irreversible adsorption (Figs 11 and 12).

Our new isolates of small siphoviruses in genera *Kagunavirus*, *Skatevirus*, and others encode tailspikes for recognition and enzymatic processing of the O16-type O-antigen that are encoded directly downstream of a J-like central tail fiber protein (S2 Fig). Notably, this is the same locus where the *bona fide* RBPs described in our previous work are encoded [10] as well as the capsule-targeting tailspikes of small siphoviruses targeting *Klebsiella* spp. (S2 Fig) [113]. We therefore conclude that these different RBPs are used analogously for terminal receptor recognition. Interestingly, the tailspikes of these distinct siphoviruses as well as *Kayfunavirus* ElsieAttenhofer (Bas105), *Gamaleyavirus* RudolfBernhard (Bas96), and *Lederbergvirus* CurroJimenez (Bas106) are all homologous and feature an enzymatic pectin lyase-like receptor-binding domain (InterPro accession IPR011050 [114]) connected to variable anchor domains (Figs 11A and 12A). A phylogeny including distant relatives known to target the O:4 O-antigen of *Salmonella* Typhimurium such as P22 shows that the tailspikes of our O16-targeting phages form a distinct, though diverse, group alongside tailspikes from a few phages isolated in previous studies (Fig 11A) [64]. This close relationship between tailspikes from very different phages targeting O16-type O-antigen highlights the frequent horizontal exchange of loci encoding these RBPs [33,115].

Based on prominent examples of glycan-targeting tailspikes, we had initially expected that phages depending on O-antigen as host receptor would mostly use enzymatic tailspikes to penetrate the glycan barrier [33,78,94]. However, many of our O16-dependent phage isolates have tail fibers instead. While these vary in length and the types of receptor-binding domains (RBDs), many contain an RBD homologous to that of Gp12 short-tail fibers of T-even phages targeting the *E. coli* LPS core [116] (InterPro accession SSF88874 [114]). These Gp12-like RBDs are located in the distal parts of lateral tail fibers across various phages, including myoviruses of the *Wifcevirus* genus, siphovirus IrisVonRoten (Bas32), and podoviruses of the *Gordonclarkvirinae* subfamily (Fig 11B). However, such a broad distribution of closely related RBDs is not unexpected, given the frequent horizontal transfer of tail fiber components [115].

Interestingly, a phylogeny shows that the Gp12-like RBDs of *Wifcevirus* isolates targeting O16-type O-antigen are scattered between homologs from diverse other *Wifcevirus* phages, suggesting that they either mediate very broad host recognition or that these other *Wifcevirus* isolates could also target this specific glycan (Fig 11B). Conversely Gp12-like RBDs of *Gordonclarkvirinae* targeting O16-type O-antigen are only very distantly related to homologous domains of other phages from this group (Fig 11B). For one of these others, EP335, it had been shown previously that its lateral tail fiber directly binds O157-type O-antigen, suggesting that O-antigen recognition may be a conserved feature of these tail fibers with Gp12-like RBDs [117]. Other phages from the expanded BASEL collection encode different types of tail

fibers with other RBDs that all likely target this glycan, highlighting the versatility of these non-enzymatic RBPs (see predicted structures in Figs 12B, 12C, and S7).

## Phages with several receptor-binding proteins or dual receptor-binding domains

Notably, some of our new isolates that depend on O16-tpe O-antigen of *E. coli* K-12 encode two and not only one *bona fide* RBP. Of these, *Jilinvirus* HeidiAbel (Bas97) encodes separate long and short-tail fibers that are likely used successively during host recognition analogous to the long and short-tail fibers of much larger T-even myoviruses [10,73,74] (S7 Fig). A very different situation is apparent for *Gamaleyavirus* RudolfBernhard (Bas96), which encodes two tailspikes of which one carries a pectin lyase-like domain closely related to others from our collection (Figs 11A and 12A) and the other one seems unrelated. Such a setup of two tail-spikes is common for *Gamaleyavirus* phages and other *Schitoviridae* (see also S8 Fig) [33,78]. It seems likely that the two tailspikes are used for alternative target recognition similar to *Autographiviridae* phage SP6, which carries two separate tailspikes that it can use to target different types of O-antigen to infect different strains of *Salmonella* [94].

While the presentation of two independent RBPs is thus not uncommon for podoviruses, the *Gordonclarkvirinae* are the only phages of this morphotype in the BASEL collection that encode separate long- and short-tail fibers, which are likely used successively and not alternatively [57]. The short-tail fibers of these phages are highly conserved and appear to be vertically inherited (Figs 12C and S9A) [57]. Similarly, the *Wifcevirus* phages also feature two different tail fibers, of which one—predicted to have an elongated RBD tip similar to the T4 Gp37 long-tail fiber [37] (Fig 12C)—is highly conserved across this genus (S9B Fig). We therefore suggest that these conserved RBPs have a more mechanical role during the infection process rather than recognizing divergent host receptors. Conversely, both the *Gordonclarkvirinae* and *Wifcevirus* phages feature a second tail fiber with a predicted distinctive T-shaped structure presenting dual RBDs, the Gp12-like RBD1 (see above) and a second, T5-like RBD2 that could not be identified from primary sequence alone (Figs 12B and S10A–S10C). Both the Gp12-like RBD1 and the T5-like RBD2 differ widely between homologs of the T-shaped tail fibers (S9C Fig) and have a phylogenetic distribution that is scattered across viral groups and morphotypes (Figs 11B and S10D). These observations are in line with the recognition of highly variable host receptors like O-antigen, suggesting that these T-shaped tail fibers may generally target these glycans and possibly others like capsules.

This remarkable T-shaped tail fiber structure is predicted to have a very similar arrangement in *Gordonclarkvirinae* (exemplified by Gp10 of AnnaHegner, Bas87), *Wifcevirus* (exemplified by Gp114 of ManiWeber, Bas99) and also the lateral tail fibers of some *Tequintavirus* phages like IrisVonRoten (Gp24, Bas32; see Fig 12B). Using Gp24 of IrisvonRoten as a representative, the DALI server identified structural similarities and shared structural elements between the Gp12-like RBD1 with X-ray crystal structures of the distal knob domain (D10) of the T4 long-tail fiber Gp37 [118] (Z-score 8.8, RMSD 3.2, 100 residues) and—as expected from primary sequence—with the Gp12 short-tail fiber distal tip [118] (Z-score 6.2, RMSD 4.1, 127 residues; S10A–S10C Fig). Likewise, the T5-like RBD2 of Gp24 of IrisVonRoten also shares structural similarity with the distal binding tip and intramolecular chaperone (IMC) domain of the phage T5 lateral tail fiber [119] (Z-score 6.3, RMSD 4.8, 179 residues; see S10A–S10C Fig). Finally, while the T-shaped RBPs of IrisVonRoten (Gp24) and AnnaHegner (Gp10) connect to a long proximal fiber (Gp23 and Gp9, respectively), the T-shaped RBP of ManiWeber (Gp114) instead features a prominent attachment domain at its N-terminus that likely facilitates direct baseplate attachment (Fig 12B). This diversity highlights the modular versatility of bacteriophage RBPs for host recognition and the initiation of viral infections.

## Host range across pathogenic *E. coli* and *Salmonella* Typhimurium

The host range of bacteriophages is a crucial property both with regard to their ecology and for applications in clinics and biotechnology [120–122]. It can be defined in different ways, but researchers generally distinguish a phage's lytic host range—estimated by the ability to lyse a host strain in a spot test—and its plating host range, i.e., the ability to form plaques [120,121]. While plaque formation requires the phage to bind the host cell and to overcome or bypass all layers of bacterial immunity at least partially, the lytic host range is primarily informative about viral host recognition due to "lysis from without" at high phage-to-host ratios even in the absence of viral replication [123]. In our previous work, we tested the host range of BASEL phages and classical model phages against commonly used model strains of *E. coli* and *S.* Typhimurium [10]. For the current study, we anticipated that the diverse O-antigen variants of these strains would largely exclude the mostly O16-dependent phages presented here. This would make it impossible to estimate their interaction with bacterial immunity. We therefore included three additional *E. coli* strains encoding O16-type O-antigen that belong to sequence types (ST)-131, −144, and −8,281 (see section *Bacterial strains and strain construction* in Materials and methods).

Our experiments to determine the lysis host range indeed revealed that most O16-dependent phages lysed all or some of the O16-expressing strains but barely any of the strains with rough LPS or another type of O-antigen (Fig 13A). This is in line with the expected dominant role of O-antigen for the host recognition of these phages, especially since most of them do not encode additional receptor-binding proteins besides the O16-targeting tailspikes or tail fibers (Figs 11 and 12) [29–31,35]. However, there were stark differences between the different groups of phages that we tested. All small siphoviruses (genera *Kagunavirus*, *Skatevirus*, and relatives; Bas70–82) and many podoviruses (Bas95/96 Bas102–106) had a very narrow host range beyond strains encoding O16-type O-antigen and lysed almost none of the strains without it (Fig 13A). These phages have either an O16-targeting enzymatic tailspike or simple tail fibers (Figs 11, 12 and S7), suggesting that their ability to target additional host receptors is limited. Conversely, isolates of *Gordonclarkvirinae* and *Wifcevirus* groups showed a very broad host recognition and could in some cases even lyse strains with rough LPS (Fig 13A). This indicates that the T-shaped tail fibers of these phages are more versatile and might be able to bind different glycan structures with their two RBDs (Figs 11, 12, S7, S9B, S9C, and S10). Previous work had indeed already observed a comparably broad recognition of some isolates of large myoviruses (e.g., *Tevenvirinae*, *Vequintavirinae* and relatives, etc.) or large siphoviruses (e.g., *Markadamsvirinae*) that commonly carry complex long-tail fibers [10,35,59,124,125]. Our data do not show unambiguously which of the two RBDs of the T-shaped tail fibers—or both—may be critical for O-antigen recognition and bacterial infection in our experiments. However, the phylogeny of the T5-like RBD2 does not correlate with any host recognition data (compare Figs 13A and S10D). Conversely, the phylogeny of the Gp12-like RBD1 shows different clusters for *Gordonclarkvirinae* with broadest host recognition (Bas87 and Bas91) and others for those with narrower host range (Bas86/89/94 and Bas88/89/82/83; compare Figs 11B and 13A). This may suggest a bigger role for the Gp12-like RBD in host recognition of *Gordonclarkvirinae* across our set of host strains.

The siphoviruses with O16-targeting lateral tail fibers and additional central fibers to bind a porin as terminal receptor—new *Dhillonvirus* isolates Bas83–85 as well as Bas07 (JakobBernoulli) and Bas32 (IrisVonRoten)—lysed all rough strains but differed in their ability to infect others (Fig 13A). This confirms the barrier function of mismatched O-antigen types for these phages and the expected facultative nature of O-antigen binding when the terminal receptor on the cell surface is accessible [10,28]. As shown already previously [10,12,17], phages that

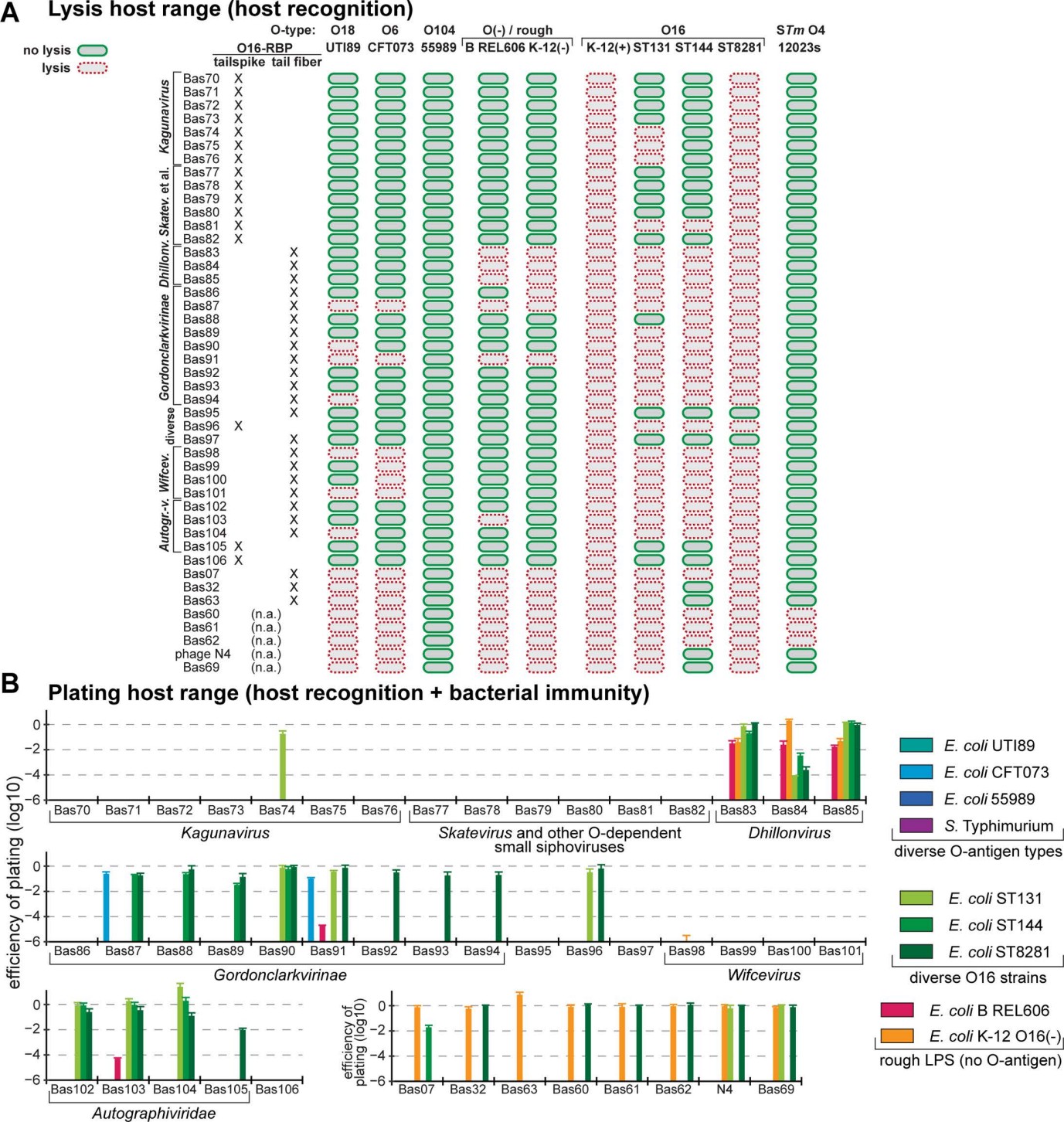

**Fig 13. Host range of new phage isolates in the BASEL collection. (A)** Lysis host range of all new BASEL phages and controls (same as in S1B Fig) as determined by qualitative spot test assays with high-titer phage stocks (>10⁹ pfu/ml if possible; see Materials and methods). The O-antigen type of different *E. coli* and *Salmonella* Typhimurium strains and possibly rough LPS (as O(-)) are highlighted at the top, while phages with O16-targeting tail spikes or tail fibers are distinguished on the left side. Lysis of a host strain is shown by a red cell lysis icon while the inability to lyse a strain is shown as a green icon in the shape of a cell with full integrity. **(B)** Plating host range of all new BASEL phages and controls (like in (A)) as determined by quantitative efficiency-of-plating (EOP) assays using serial dilutions of phage stocks. Data points and error bars represent average and standard deviation of at least three independent experiments. Raw data and calculations are available in S1 Data. In (A) and (B), K-12 strains have been selected differently for phages inhibited by its cryptic prophages (Δ9CP) and for those who are not (ΔRM).

can bypass O-antigen by targeting the NGR glycan (Bas60–62 and N4 as well as Bas69) have a very broad host recognition with the *Justusliebigvirus* phages Bas60–62 featuring the largest lysis host range which even includes *Salmonella* Typhimurium (Fig 13A). This is likely due to the array of different tailspikes and tail fibers expressed by these phages, which enables them to overcome diverse glycan barriers [126]. Conversely, we also observed that many apparently O16-dependent phages failed to lyse one or more of the strains with this O-antigen type that we tested, especially when the phages use enzymatic tailspikes for host recognition (Fig 13A). Since these phages do not encode any other receptor-binding proteins, we find it unlikely that bacterial variation in other potential receptors such as the LPS core may be responsible for this behavior. Instead, it seems likely to us that strain-specific O-antigen modifications—which are abundant and diverse ([recently reviewed in reference 36])—could prevent the binding of some highly specific tailspikes to certain variants of O16-type O-antigen.

The plating host range of all phages was usually much narrower than their lysis host range (Fig 13A and 13B), which shows the impact of bacterial immunity in the diverse strains that we used. This effect was particularly strong for all phages with considerable sensitivity to RM systems such as small siphoviruses Bas70–82 (Fig 13B; compare Figs 5D and 6G). Conversely, phages with pronounced resistance to RM systems of the *Gordonclarkvirinae* (Bas86-93) and *Gamaleyavirus* (Bas96) groups could form plaques on many of the strains that they can lyse, indicating that this abundant and broadly targeting form of defense systems has a major impact on phage host range (Fig 13B; compare Figs 7D and 8G) [40,46,47]. This is in line with previous work showing a broad plating host range for individual isolates of *Gordonclarkvirinae* [117,121,127] and a narrower one for individual *Wifcevirus* phages or small siphoviruses like *Kagunavirus* isolates [35,59]. Intriguingly, *Autographiviridae* phages (Bas102–105) and in particular the new *Dhillonvirus* isolates (Bas83–85) formed plaques on nearly all strains that they could lyse despite their only moderate resistance to RM systems (Fig 13A and 13B; compare Figs 6G and 10D). It would be interesting to study the underlying molecular mechanisms that apparently enable these phages to overcome the native RM systems of these strains while the *Wifcevirus* phages—displaying considerable resistance to restriction in the K-12 host (Fig 9E)—fail to form plaques on any of the many strains which they can lyse (Fig 13A and 13B).

## Cryptic prophages inhibit some BASEL phages via O-antigen glycosylation

One of the most curious biological results of this study has been the observation that phages belonging to very different viral groups—Bas74–76 (*Kagunavirus*), Bas83–85 (*Dhillonvirus*), Bas87 + Bas91 (*Gordonclarkvirinae*), and Bas98-Bas101 (*Wifcevirus*)—are inhibited by the presence of the cryptic prophages of *E. coli* K-12 (see Figs 5D, 6G, 7D, and 9E). Based on the diverse *bona fide* antiphage defense systems encoded in these genomic islands (Fig 4D), we had initially assumed that one or more of these would target the phages and impair their replication. The phylogenies of these phages show that sensitivity to inhibition by the cryptic prophages does not correlate with certain clades of their respective viral groups (Figs 5C, 6F, 7C and 9B). This observation suggested that sensitivity to this inhibition must be due to variable genes of the susceptible phage groups that could, e.g., be sensed/ targeted by bacterial immunity or may serve as antidefense factors, as shown in other cases [40,45,128]. However, we also realized that the loci encoding the receptor-binding proteins of the phages sensitive to inhibition by the cryptic prophages were distinct from their insensitive relatives. This can be seen, e.g., for the *Kagunavirus* tailspikes in Fig 11A where those of the isolates inhibited by cryptic prophages (Bas74–76) cluster in a completely different clade than the insensitive ones (Bas70–73). Similarly, the only features specifically shared by *Gordonclarkvirinae* AnnaHegner (Bas87) and LorandFenyves (Bas91), which are sensitive to the cryptic prophages, is the variant of their Gp12-like tail fiber RBD which is different from those of all other isolates (Fig 11B). We thus suspected that viral adsorption and not classical anti-phage

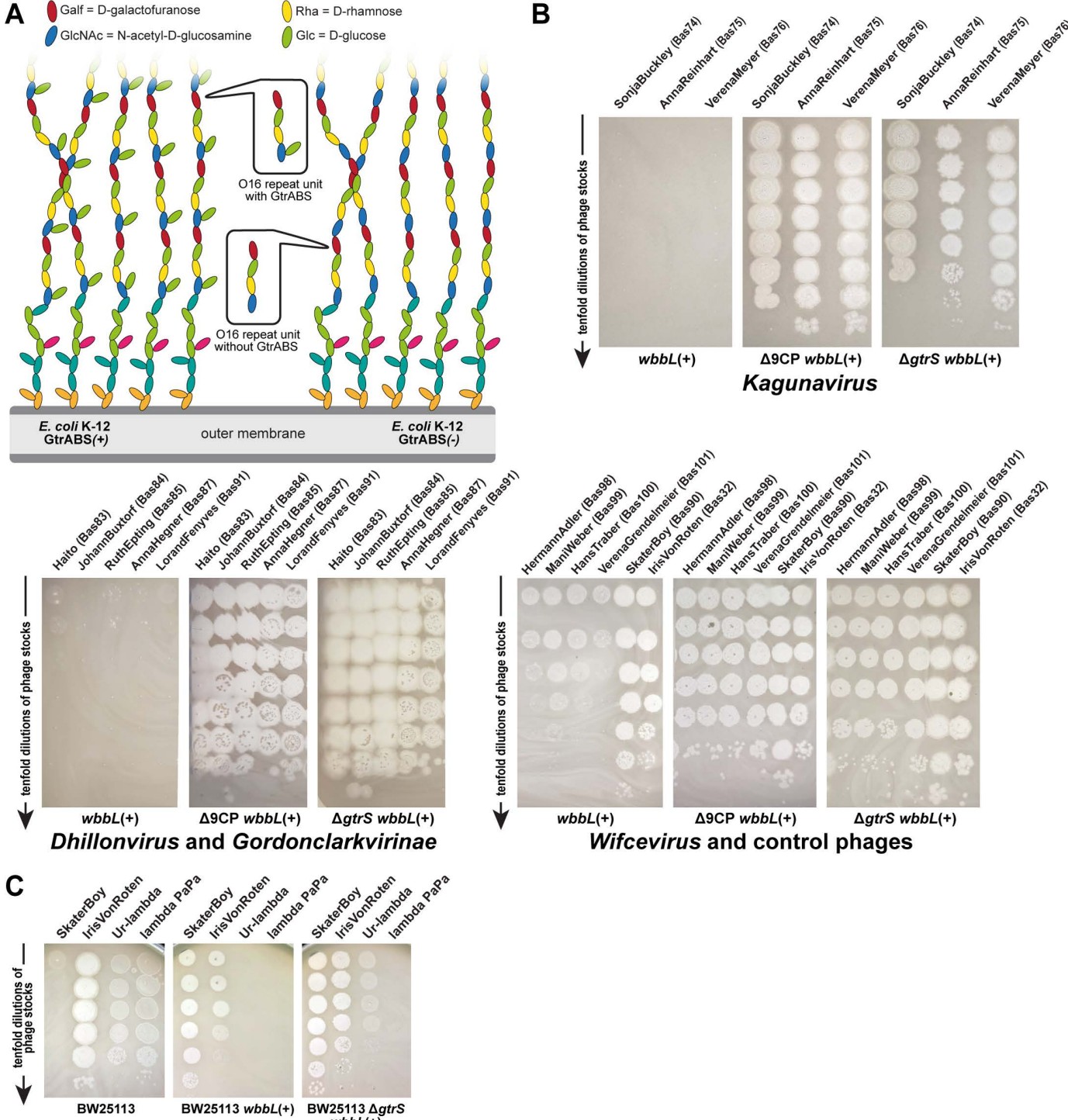

**Fig 14. GtrABS-mediated O-antigen glycosylation inhibits phage infection.** **(A)** O16-type O-antigen of *E. coli* K-12 with (left) or without (right) the GtrABS system of CPS-53 differs in the presence or absence of the glucose side chain, respectively [129,130]. **(B)** Serial dilutions of different phages were spotted on variants of *E. coli* K-12 BW25113 with restored O-antigen expression as well as deletion of the GtrS glucosyltransferase (Δ*gtrS*) or the nine cryptic prophages including CPS-53 (Δ9CP). Three *Dhillonvirus* phages (Bas83–85), two *Gordonclarkvirinae* (Bas87 + Bas91), and the *Wifcevirus* phages (Bas98–101) are inhibited by the K-12 cryptic prophages if the GtrABS system in CPS-53 is intact (see S11 Fig for the same experiment without O-antigen expression). *Gordonclarkvirinae* phage SkaterBoy (Bas90) and *Tequintavirus* IrisVonRoten (Bas32) were used as GtrABS-insensitive controls that do or do not depend on O16-type O-antigen, respectively. A small growth defect of the three *Kagunavirus* phages in the Δ*gtrS* mutant compared to the Δ9CP strain (top right) is indicative of a slight inhibition of these phages by other factor(s) encoded in the cryptic prophages **(C)** Serial dilutions of lambda PaPa and Ur-lambda were spotted on variants of *E. coli*

K-12 BW25113 either in its wildtype form (left) or with restored O-antigen expression and either active (middle) or inactivated (right) GtrABS system. Phages SkaterBoy (Bas90, *Gordonclarkvirinae*) and IrisVonRoten (Bas32, *Tequintavirus*) were added as controls that do or do not depend on the O16-type O-antigen and are both insensitive to GtrABS.

defense systems could be responsible at least for some of the observed differences in sensitivity to the presence of cryptic prophages.

Interestingly, the cryptic prophage CPS-53 (also known as KplE1) of *E. coli* K-12 encodes a GtrABS O-antigen glycosylation system that adds the glucose side chain to the O16-type O-antigen of this strain for which its O-antigen biosynthesis locus itself lacks a dedicated glycosyltransferase (Fig 14A) [129,130]. Consistently, this side chain could not be found when the O-antigen structure of another *E. coli* strain with O16-type O-antigen was solved [131]. Given that O-antigen modifications can impair the adsorption of bacteriophages [36], we tested the role of the CPS-53 GtrABS system for the host range of our phage isolates that were inhibited by the K-12 cryptic prophages. Remarkably, genetic inactivation of this system resulted in susceptibility of *E. coli* K-12 to all these phages, suggesting that O-antigen glycosylation and no other aspect of bacterial immunity had inhibited their growth on *E. coli* K-12 with cryptic prophages (Fig 14B). These results were reminiscent of previous work showing that homologous tailspikes of phages EP75 and HK620 target differently glucosylated variants of *E. coli* O18 O-antigen, possibly due to differences in glycan repeat recognition [117]. Unlike the other phages that depend on O-antigen as host receptor, the *Dhillonvirus* isolates Bas83−85 were only affected by the GtrABS system in presence of restored O-antigen expression while they infected *E. coli* K-12 with and without cryptic prophages equally well in the absence of this glycan barrier (Figs 14B and S11). The *Dhillonvirus* phages have two RBPs and do not need recognition of the O-antigen as primary receptor to target their terminal porin receptor, but they depend on O-antigen recognition—which is apparently impaired by GtrABS glycosylation—to cross this layer when it is present [10,100].

These observations with the *Dhillonvirus* phages were reminiscent of the conundrum that phage lambda—initially found as a prophage in *E. coli* K-12—was unable to infect this strain after restoration of its O-antigen [10]. No plaque formation in the presence of this glycan barrier could be seen both with the commonly used laboratory variant lambda PaPa or the original Ur-lambda (Fig 14C) [10,132]. Unlike the domesticated lambda PaPa, Ur-lambda does not only feature a central tail fiber targeting the LamB porin as a terminal receptor but also side tail fibers suggested to bind OmpC as the primary receptor [132]. Our results raised the question how lambda would initially have lysogenized the original *E. coli* K-12 strain that expressed O16-type O-antigen [27]. Interestingly, we find that Ur-lambda with its side tail fibers—but not lambda PaPa without them—can overcome the O16-type O-antigen of *E. coli* K-12 in the absence of GtrABS-mediated glycosylation (Fig 14C). These results suggest that the side tail fibers of Ur-lambda can bind unmodified but not glucosylated O16-type O-antigen as host receptor similar to the recognition of O-antigen as a primary receptor by other siphoviruses like JakobBernoulli (Bas07) or IrisVonRoten (Bas32; see Fig 1) [10]. Phage lambda might thus have lysogenized *E. coli* K-12 either before acquisition of CPS-53 or while the GtrABS system was transiently not expressed since related systems, e.g., in *Salmonella*, are controlled by phase variation [133].

## Discussion

### Phages depending on O-antigen glycans are diverse and abundant in nature

While it is curious that we find O16-targeting phages to be much more abundant in environmental samples compared to sewage treatment plant inflow (S1 Table), previous work has also

observed such differences in other contexts. As an example, phages targeting the conjugative pili of different enterobacterial plasmids were much more common in wastewater compared to river water [134]. While the underlying ecological phenomena remain elusive, it seems certain that wastewater does not merely contain a large abundance of *E. coli* phages compared to most environmental samples but also has its own biases.

One interesting observation for many O16-dependent phages characterized in this work is that they seem to use these glycans as terminal host receptors, while their genomes lack any recognizable strategy to inactivate this receptor during infections. Conversely, virulent phages targeting porins characteristically encode proteins such as Cor of T1 or Llp of T5 that disable their porin receptors to prevent adsorption of viral offspring to the debris of the lysed virocell [135,136]. Similar mechanisms are also common among temperate phages, including those targeting O-antigen glycans, which they can chemically modify, e.g., by glycosylation or acetylation ([recently reviewed in reference 36]), for "superinfection exclusion" of related viruses [14,137]. These examples show that such exclusion mechanisms would in principle also be possible for virulent phages, but it seems likely that the abundance of O-antigen molecules on the cell surface (e.g., compared to porins) makes a complete modification during lytic infections futile. Instead, recent work suggested an alternative strategy of phages that transiently block their offspring's host recognition by "caps" made of chaperones on receptor-binding proteins instead of disabling the host receptors [138]. We expect that this strategy and similar concepts may be more widespread among glycan-targeting phages, and we look forward to future research uncovering their molecular mechanisms and evolutionary context.

Traditionally, siphoviruses infecting *E. coli* were considered to exclusively target porins as terminal host receptors like well-studied model phages T1 or T5 [60,61]. However, more recent studies had already described siphoviruses infecting *E. coli* or other enterobacteria that target flagella, conjugative pili, or surface glycans as host receptors [33,93,139,140]. Using *E. coli* K-12 with restored O16-type O-antigen expression as host, we indeed isolated diverse small siphoviruses from genera *Kagunavirus*, *Skatevirus*, and others that use this surface glycan as an essential host receptor. It seems likely that the abundance and diversity of these groups had previously been overlooked due to the traditional focus on *E. coli* laboratory strains with rough LPS (see Fig 1A). Our results strongly suggest that O-antigen is the only and terminal host receptor of these isolates because they only carry a single receptor-binding protein—in all cases enzymatic tailspikes (Figs 11A and 12A)—which is encoded directly next to the J-like central tail fiber hub (S2A Fig). This is the same locus where other small siphoviruses infecting *E. coli* encode *bona fide* receptor-binding proteins as well as superinfection exclusion factors targeting different porins (S2B Fig [10,135]), and host recognition by these central tail fibers is known to trigger irreversible adsorption and genome injection [110,141]. We look forward to future studies characterizing the mechanistic link of O-antigen recognition and genome injection in more detail. One specific aspect deserving further investigation is the way how these processes are linked on the molecular level compared to phages with two distinct tail fibers carrying receptor-binding domains. Electron microscopy of small siphoviruses depending on O-antigen clearly shows three distal tail appendages that are likely the tailspikes presented on the J-like fibers (see Figs 5B and 6C). Conversely, siphoviruses targeting porins generally have only one central tail fiber targeting this terminal receptor besides three lateral tail fibers targeting O-antigen or other initial receptors [100]. This indicates that the single RBPs of siphoviruses targeting only O-antigen may be presented on the virion like lateral tail fibers while being functionally equivalent to central tail fibers for irreversible adsorption and triggering DNA genome ejection.

Our study highlighted once more that—besides a few models like P22, G7C, and T4 [37,72,78,142]—we lack a molecular understanding how many phages use their diverse tail

fibers and tailspikes to recognize host receptors and initiate infections. While we had initially expected to isolate many phages with tailspikes targeting O16-type O-antigen that would move the virion to the cell surface by enzymatic activity, many instead carried tail fibers lacking enzymatic properties (Figs 12B–12C and S7). Beyond a mere binding to their LPS receptor (which has been well studied for several examples [37,70,119]), we speculate that successive transient interactions of tail fibers with O-antigen receptors may enable the fibers to "glide" through this barrier towards the cell surface (see Figs 7A and 9A). Of particular interest are the novel T-shaped tail fibers with separate Gp12-like and T5-like RBDs (Figs 12B and S10) that—to the best of our knowledge—are the first example of a tail fiber protein with two distinct RBDs in the same polypeptide. Conversely, branched structures combining distinct RBPs have been well described [143], and the long-tail fiber tip Gp37 of phage T4 features different patches in the same RBD to target distinct receptors [37].

## Phage host range, glycan receptors, and bacterial immunity

Understanding the molecular mechanisms underlying the host range of different bacteriophages is important not only for ecological insights but also to enable effective applications in clinics [35,54–56,122]. While diverse aspects of phage–host interactions like bacterial metabolism and antiviral immunity all influence phage host range, the access to and recognition of viral host receptors generally play major roles [144,145]. As expected from previous work [10,28,30,35], we observed a dominant role of the O-antigen barrier for phage host range in a way that O-antigen types mismatched to the phages' receptor-binding proteins abolished host recognition while phages with no other host receptor also fully depended on its presence. Interestingly, our data show that phages with tail fibers tend to have a broader host recognition than phages with tailspikes among the tested viruses (Fig 13A). Understanding these differences on the molecular level by studying distinct functionalities of these different types of RBPs more systematically would not only expand our knowledge of phage–host interactions but could also help applications in clinics and biotechnology, e.g., by phage host range engineering [146,147]. The expanded BASEL collection with diverse phages that target the same receptor—O16-type O-antigen—could be a useful tool to investigate these interactions.

The dual role of O-antigen as a barrier and as a host receptor is reminiscent, e.g., of the well-studied capsules of *Klebsiella* spp., which form a robust barrier to phage infection but also serve as a dominant factor for phage host range by specific recognition of capsule types by certain variants of RBPs [113,148,149]. The specificity and dominance of these interactions even enable their use for quite reliable phage host range predictions across *Klebsiella* strains [150]. Conversely, for other hosts like *P. aeruginosa,* the availability of long retractable type IV pili reduces the importance of glycans as host receptors, and for many Gram-positives like *Staphylococcus aureus* at least the host recognition is usually broader due to a lower inter-strain diversity of bacterial surface structures [14,151–153].

For *E. coli*, the picture is certainly more nuanced because, e.g., most of the O16-targeting phages that can lyse our different tested strains with this O-antigen type fail to form plaques on them (compare Fig 13A and 13B). While this shows a clear influence of bacterial immunity, a recent study that experimentally tested the host range of many *E. coli* phages across many strains did not observe a simple correlation between phage host range and the number of predicted antiviral defense systems [35], unlike what had been observed for *P. aeruginosa* and *Vibrio* spp. [16,154,155]. In agreement with recent work by David, Plantady, and colleagues on a different strain of *E. coli* [145], our results primarily suggest a link between the ability to form plaques on diverse strains and viral resistance to RM systems as the most common and most broadly targeting form of bacterial immunity (see above) [46,47,156].

However, the relative importance of different antiviral defense systems for observed phage host range would certainly be a promising field for future study beyond mere predictions, e.g., by contrasting the phage susceptibility of wildtype hosts with mutants specifically lacking one or more defense systems. Our results indicate that—after removal of restriction systems in our previous work [10]—the *E. coli* K-12 genomic background may not encode anti-phage defense systems in narrow sense that play a major role in phage host range. Even when all predictable defense systems were absent in the Δ9CP strain lacking all cryptic prophages (Fig 4D; see *Bacterial strains* in Materials and methods), the new viruses that could be isolated were merely susceptible to an O-antigen glycosylation system encoded in one of these genomic islands (Fig 14).

## Expanding the BASEL collection to cover all major groups of *E. coli* phages

The value of the BASEL collection as a tool for molecular biology research lies in its assorted diversity across and within phage groups and on their ability to infect the model organism *E. coli* K-12 [6]. In this study we therefore aimed at enhancing the assorted diversity to cover all known major groups of *E. coli* phages without a major compromise on domesticated *E. coli* K-12 as host strain (Fig 15). After restoring its O-antigen and inactivating all predicted antiviral defenses—of which only O-antigen glycosylation played a detectable role –we indeed readily isolated representatives of diverse previously inaccessible viral groups, most of which strictly depend on O-antigen as host receptor. These include, e.g., diverse groups of small siphoviruses like the *Kagunavirus* genus, myoviruses of the *Wifcevirus* genus, and podoviruses of *Gordonclarkvirinae* or genus *Kayfunavirus*. Since phages from most of these diverse groups had never been isolated with any *E. coli* lab strain, there were only scarce—if any—previous studies on their molecular biology. This limitation restricted our analysis of the genomic and phenotypic data for some of them largely to data from the current study, while in our previous work on more well-studied groups of phages we could draw heavily from the generations of previous research in the field [10]. To improve the genome annotation of our new phage isolates, we therefore created a new tool—GAPS (Gene Annotation Pipeline based on sequence and structure Similarity; see Materials and methods) to infer functional information from remote homologies and structure predictions.

We anticipate that the completed BASEL collection will support various applications, from fundamental research on viral ecology and evolution over the work on new antiviral defense systems to studies on phage therapy. With an expanded biological diversity in still manageable numbers, these phages can all be grown on a regular *E. coli* K-12 by using a strain with inactivated GtrABS system in two variants, once with and once without restored O-antigen expression. Given the increasing interest in phage therapy but also its notable incidence of treatment failure [85,157,158], it is clear that not only fundamental research but also clinical applications could profit from a better understanding of links between phage genomes and taxonomy on one side and relevant biology/ phenotypes + clinical properties on the other side [35,54–56]. This is becoming increasingly evident as more and more engineered phages for therapeutic applications are developed [159]. For these applications, phages are mostly used as a vehicle to deliver proteins from nonviral origin such as CRISPR-Cas nucleases or antibacterial toxins [62,159–162]. This limitation highlights our lack of knowledge about the diverse molecular biology of phage–host interactions beyond a few traditional models. Unraveling the molecular mechanisms underlying new aspects of phage–host interactions—e.g., with tools like the BASEL collection or modern biochemistry and structural biology—could thus open up the untapped potential of engineered phages for applications in clinics or biotechnology.

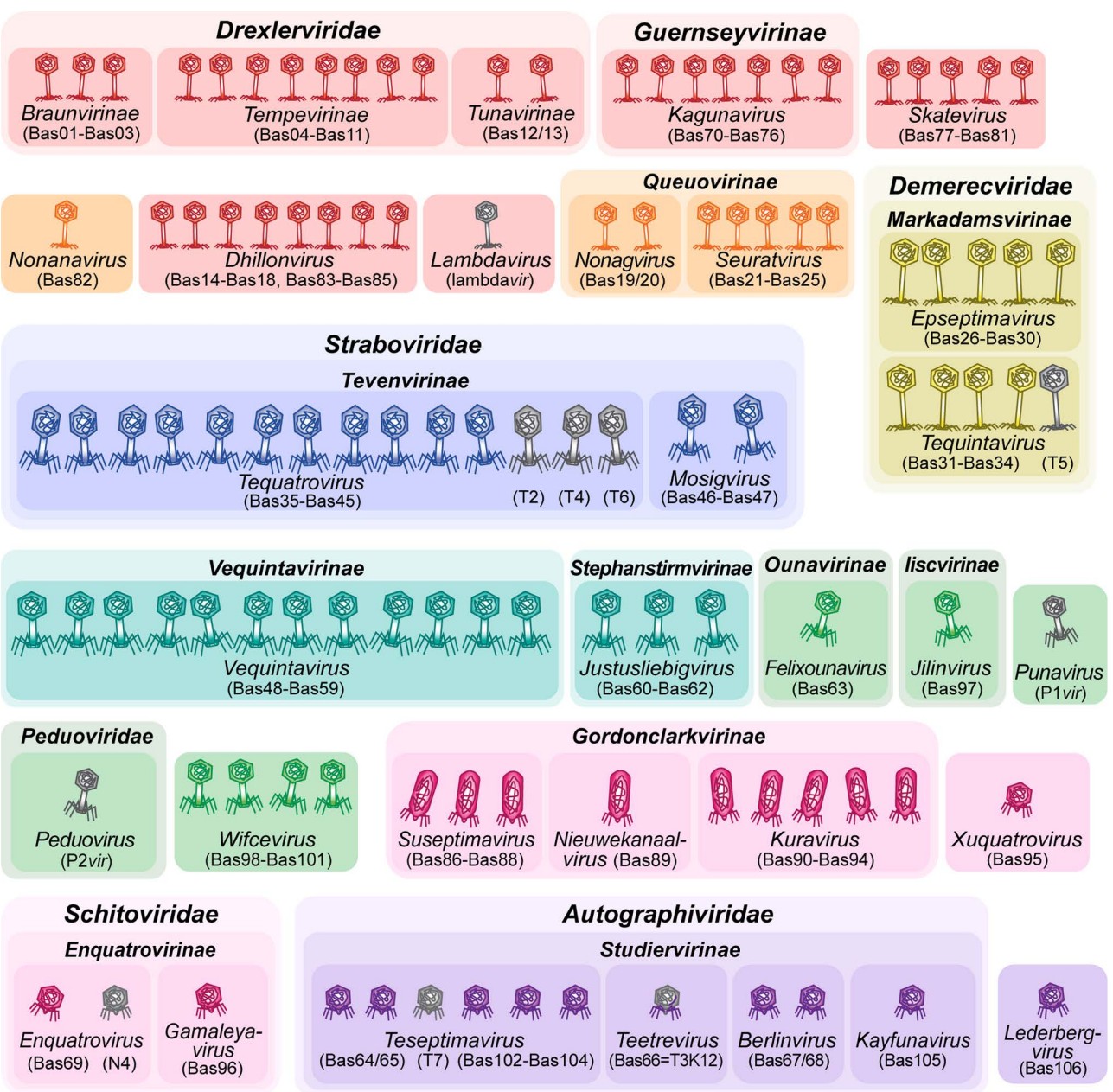

**Fig 15. Overview of the complete BASEL collection.** The illustration shows all isolates of the complete BASEL collection with their Bas## identifiers. Isolates are sorted by morphotype and their current ICTV classification [68]. The color code distinguishes phages based on morphotype and genome size. For siphoviruses, small phages (ca. 30–50 kb) are shown in red, medium-sized phages (ca. 50–100 kb) are shown in orange, and large phages (>100 kb) are shown in yellow. For myoviruses, small phages (ca. 30–100 kb) are shown in green, medium-sized phages (ca. 100–150kb) are shown in cyan, and large phages (>100kb) are shown in blue. For podoviruses, small phages (ca. 30–50 kb) are shown in violet, and medium-sized phages (ca. 50–100 kb) are shown in pink.

# Materials and methods

## Preparation of culture media and solutions

Lysogeny Broth (LB) was prepared by dissolving 10 g/l tryptone, 5 g/l yeast extract, and 10 g/l sodium chloride in Milli-Q H$_2$O and then sterilized by autoclaving. LB agar plates were

prepared by supplementing LB medium with agar at 1.5% w/v before autoclaving. Phosphate-buffered saline (PBS) was prepared as a solution containing 8 g/l NaCl, 0.2 g/l KCl, 1.44 g/l $NA_2HPO_4 \cdot 2H_2O$, and 0.24 g/l $KH_2PO_4$ with the pH adjusted to 7.4 using 10 M NaOH and sterilized by autoclaving. SM buffer was prepared as 0.1 M NaCl, 10 mM $MgSO_4$, and 0.05 M Tris (pH 7.5) and sterilized by autoclaving.

## Bacterial handling and culturing

*Escherichia coli* and *Salmonella enterica* subsp. *enterica* serovar Typhimurium strains were routinely cultured in LB medium at 37 °C in glass tubes, plastic tubes, or Erlenmeyer flasks with agitation at 170 rpm. Selection for genetic modifications or plasmid maintenance was performed with ampicillin at 50 μg/ml, kanamycin at 25 μg/ml, and zeocin at 50 μg/ml.

## Bacteriophage handling and culturing

Bacteriophages were generally cultured using the double-agar overlay method [163] with improvements of classical techniques described by Kauffman and Polz [164]. Top agar was prepared as LB agar with only 0.5% w/v agar supplemented with 20 mM $MgSO_4$ and 5 mM $CaCl_2$. Top agar plates were incubated at 37°C and plaques were counted as soon as they were identified by visual inspection. However, the plates were always incubated for at least 24 hours to record also slow-growing plaques.

High-titer stocks of bacteriophages were generated using plate-grown viruses. Briefly, phages were spread with a loop throughout the top agar before solidification to achieve confluent lysis after incubation for 16–24 hours at 37°C. Subsequently, top agars were scraped off from the bottom agar into each 15 ml of SM buffer and centrifuged at 8000*g* for 10 min. Supernatants were used as high-titer stocks and stored in the dark at 4°C. Phage Ur-lambda was obtained from the supernatant of an exponentially growing culture of the *E. coli* K-12 EMG2 strain (CGSC #4,401)—carrying an Ur-lambda prophage—after 2 hours of treatment with 0.5 μg/ml of mitomycin C to trigger prophage induction [132,165].

Phages of the original BASEL collection from our previous work [10] were generally grown on *E. coli* K-12 MG1655 ΔRM. New isolates from this study were propagated on *E. coli* K-12 MG1655 ΔRM with restored O-antigen (carrying pAS001 or the *wbbL*(+) genotype) except those with sensitivity to GtrABS-mediated O-antigen glycosylation. These phages were cultured on *E. coli* K-12 BW25113 Δ9CP or *E. coli* K-12 BW25113 Δ*gtrS* (both with restored O-antigen, i.e., carrying pAS001 or the *wbbL*(+) genotype). All bacteriophages used in this study are listed in S2 Table.

## Bacterial strains and strain construction

All bacterial strains used in this work are listed in S3 Table, all oligonucleotide primers in S4 Table, and all plasmids in S5 Table.

## Variants of *Escherichia coli* K-12 MG1655 and BW25113

The primary host of our experimentation with *E. coli* K-12 was the ΔRM mutant of the MG1655 laboratory strain, which we had reported in our previous work [10]. Briefly, we had engineered this strain to be a more permissive host for bacteriophages by knocking out all known restriction systems of *E. coli* K-12, i.e., the EcoKI type I RM system (encoded by *hsdRMS*) and the McrA, Mrr, and McrBC type IV restriction systems. For phage isolation and phenotyping, the strain carried a variant of the F-plasmid in which the *pifA* immunity gene had been deleted [10]. Phenotyping with plasmid-encoded antiviral defense systems was

done with a strain carrying pBR322_ΔP*tet* as an empty vector control [10,166] (see S3 Table). Whenever indicated, we used the *E. coli* K-12 ΔRM variant with a repaired chromosomal *wbbL* gene that has restored O16-type O-antigen expression from our previous work [10] (see S3 Table). To rapidly restore O-antigen expression in diverse strains (e.g., porin mutants of the Keio single-gene knockout collection [167]), we created plasmid pAS001 that constitutively expresses *wbbL* (see below in section *Plasmid construction*). Fortuitously, the plasmid-encoded *wbbL* does not fully restore the O-antigen barrier (because most phages inhibited by it still show infectivity; compare Figs 1 and S11A) but enough of it to enable robust infections by all O16-dependent phages (compare Figs 2 and S11B).

To identify possible limitations of phage infectivity beyond host recognition, we focused on possible antiviral defenses in the K-12 genomic background beyond the restriction systems knocked out in *E. coli* K-12 MG1655 ΔRM. One of these is the enigmatic enterobacterial type I-E CRISPR-Cas system known to be not functional in phage defense without dedicated genetic engineering [168,169]. In addition, *E. coli* K-12 strains encode Lit (reported to cleave bacterial EF-Tu when sensing the major capsid protein of some *Tevenvirinae* [170]), RnlAB (toxin-antitoxin system that aborts T4 infections in absence of the viral *dmd* "master key" antitoxin [171]), DicB (inhibitor of the bacterial ManYZ inner membrane conduit of lambda phage DNA translocation [172]), and a Hachiman type I system (targeting DNA replication of diverse phages; also known as AbpBA [44,47,173]). With the exception of the seemingly inactive type I-E CRISPR-Cas system, all these *bona fide* antiviral defenses are encoded in the cryptic prophages of *E. coli* K-12. We therefore used the *E. coli* K-12 BW25113 Δ9CP strain devoid of all cryptic prophages as an additional host for phage isolation and characterization [51]. This strain is derived from the BW25113 laboratory wildtype of *E. coli* K-12 which has a few minor differences compared to the MG1655 laboratory wildtype [174,175] (see also Fig 4D). Most important for our work, *E. coli* K-12 BW25113 carries active variants of the three type IV restriction systems knocked out in our MG1655 ΔRM strain as well as the *hsdRMS* locus encoding the EcoKI type I RM system. Unfortunately, due to the high sensitivity of *E. coli* K-12 BW25113 Δ9CP to stress and possibly also as a consequence of its already extensive genomic engineering, our attempted knockouts of these systems consistently failed. However, the EcoKI locus of BW25113 is restriction-deficient, and we are not aware of any wildtype phages that would be sensitive to the three type IV restriction systems of *E. coli* K-12, suggesting that the presence of these loci might have only a negligible effect. Analogous to our work with *E. coli* K-12 MG1655 ΔRM [10], we also generated a variant of *E. coli* K-12 BW25113 Δ9CP with chromosomally restored *wbbL(+)* to isolate and study phages that would depend on this glycan (see S3 Table). The different variants of *E. coli* K-12 BW25113 Δ9CP were used for phage isolation and characterization just like the ΔRM variants with plasmid-encoded antiviral defenses and a *pifA*-deficient F-plasmid (see S3 Table).

Later during our study, we found that only the GtrABS O-antigen modification system of CPS-53 was responsible for the observed inhibition of phage growth by the cryptic prophages of *E. coli* K-12 (Fig 14). We therefore additionally used an *E. coli* K-12 BW25113 Δ*gtrS* mutant as well as a derivative of this knockout with chromosomally repaired *wbbL*—restoring O-antigen expression—for different experiments shown in this manuscript (see S3 Table). Briefly, a clean deletion of *gtrS* in the *E. coli* K-12 BW25113 strain was generated by lambda red recombineering in two steps analogous to the knockouts in our previous work [10]. In the first step, *gtrS* was replaced with a double-selectable cassette encoding kanamycin resistance and sucrose sensitivity (amplified from pUA139_cat-sacB_v3 using prDP0079/80). Subsequently, the double-selectable cassette was removed by recombineering with annealed 80 nt oligonucleotides spanning the desired deletion with 40 nt on each side (prDP0081/82). Oligonucleotide primers prDP0083/84 were used to probe the success of each step of

recombineering by overspanning colony PCR over the *gtrS* gene (all listed in S4 Table). Finally, the *wbbL(+)* gene was restored by precisely removing the IS5 element disrupting this gene, as described in our previous work [10].

## Other enterobacterial strains used for bacteriophage phenotyping

To determine the host range of our new phage isolates and relevant controls, we initially used the same strains as in our previous work (see there for details [10]). Briefly, besides the *E. coli* K-12 variants with and without restored O-antigen (see above; phylogroup A, ST10, O16(+/-):H48, K-12 LPS core) these were the rough *E. coli* B REL606 laboratory strain (phylogroup A, O7(-):K5:H(-), truncated R1 LPS core), uropathogenic *E. coli* model strains UTI89 (phylogroup B2, ST95, O18:K1:H7, R1 LPS core) and CFT073 (phylogroup B2, ST73, O6:K2:H1, R1 LPS core), and enteroaggregative *E. coli* model strain 55989 (phylogroup B1, ST678, O104:H4, R3 LPS core). Similarly, we again used the common *Salmonella enterica* subsp. *enterica* serovar Typhimurium laboratory strains 12023s (also known as ATCC 14028) and SL1344 that share the *Salmonella* Typhimurium O4-type O-antigen. Because the results with these strains were indistinguishable, only those with *S.* Typhimurium 12023s are included in this manuscript. Beyond these strains already included for host range analyses in our previous work, we included three additional *E. coli* strains with O16-type O-antigen (i.e., the same as *E. coli* K-12) belonging to ST131 (Egli lab collection isolate 720834–18; phylogroup B2, O16:H5), ST144 (Egli lab collection isolate 714478–19; phylogroup B2, O16:H6), and ST8281 (Egli lab collection isolate 711043–19; phylogroup B2, O16:H5) [176].

## Plasmids and plasmid construction

For the phenotyping of phage resistance or sensitivity to diverse bacterial immunity systems, we used plasmids encoding different antiviral defenses constructed in our previous work (see Fig 4B and 4C as well as S5 Table). Plasmid pAS001 encoding *wbbL* was cloned by inserting the *wbbL* gene downstream of a constitutively expressed chloramphenicol resistance gene on a plasmid with SC101 origin of replication (see details of plasmid construction in S6 Table).

## Bacteriophage isolation

**Basic procedure.** Phages isolates presented in this study have been isolated from diverse samples between March 2020 and July 2022. Initial experiments aimed at isolating phages that would depend on O16-type O-antigen used *E. coli* K-12 MG1655 ΔRM *wbbL(+)* as host and subsequent plating on its ancestor without O-antigen to identify O16-dependent isolates. Later experiments aimed at isolating phages inhibited by presence of the K-12 cryptic prophages used *E. coli* K-12 BW25113 Δ9CP transformed with *wbbL*-expressing pAS001 as host to obtain phages irrespective of whether they depend on O-antigen or not (compare Figs 1, 2 and S12). These isolates were later plated on *E. coli* K-12 BW25113 transformed with pAS001 to screen for impaired growth in the presence of cryptic prophages and on *E. coli* K-12 BW25113 Δ9CP without pAS001 to screen for dependence on O-antigen expression. The isolation and screening procedure is summarized in Fig 2A and 2B.

We generally isolated phages by direct plating without enrichment step to avoid biases in favor of fast-growing isolates. The procedure was performed similar to the approach in our previous work [10]. Aqueous samples were directly used in batches of 50 ml, while solid samples were first agitated overnight in PBS for suspension and release of viral particles. Subsequently, we centrifuged all samples at 8′000 g for 10 min to separate viral particles—which remain in the supernatant—from larger unorganic particles or bacterial cells. Viral particles were then precipitated by supplementation with $ZnCl_2$ at a final concentration of 40 mM and

an incubation without agitation for one hour at 37°C. Subsequently, precipitates were pelleted by centrifugation at 8'000$g$ for 10 min and pellets were resuspended carefully in each 0.5–1 ml of SM buffer depending on how much volume was required for a given pellet. For plating of phage isolates, the suspensions were mixed with 500 µl of bacterial overnight culture and, after some minutes at room temperature, with 10 ml of top agar. In the case of sewage plant inflow, serial dilutions of the resuspended precipitate were used due to the usually very high amount of phages. Top agars with samples and bacterial inoculum were then poured onto a square LB agar plate (ca. 12 cm × 12 cm). After solidification, we incubated the plates at 37°C for up to 24 hours.

**Isolation of clonal bacteriophage isolates.** After incubation, phages grew as plaques—i.e., spots of clearance—scattered across the top agar that was otherwise opaque from dense bacterial growth. To screen for phages that depended on restored O16-type O-antigen expression and/ or showed impaired growth in presence of K-12 cryptic prophages, ca. 150 plaques per plate were separately picked with sterile toothpicks and copied onto freshly inoculated soft agars of hosts without O-antigen expression and/ or cryptic prophages (see *Bacterial strains* above). When O-antigen dependency and/ or inhibition by cryptic prophages were observed, the respective isolates were propagated at least three times via single plaques on top agars of their isolation host.

**Composition of the completed BASEL collection.** After phage isolation and initial screening, we sequenced the genomes of all ca. 100 isolates that depended on O16-type O-antigen or were inhibited by the K-12 cryptic prophages (see below). Based on phylogenetic analyses, a set of ca. 40 phages was then chosen based primarily on largest genomic distances and characterized phenotypically alongside 8 phages from our previous work that grew well on *E. coli* K-12 with restored O-antigen expression (see below). We then chose 37 new phage isolates for inclusion in the completed BASEL collection by excluding a few more closely related isolates that behaved very similarly (see list in S2 Table).

## Efficiency of plating assays

We quantified the infectivity of a phage on a host strain as the efficiency of plating (EOP) [177], i.e., as the ratio of its plaque formation on that host over the plaque formation on susceptible reference strains. As references, we used *E. coli* K-12 MG1655 ΔRM *wbbL*(+) or *E. coli* K-12 BW251139 Δ9CP *wbbL*(+), depending on the isolates' sensitivity to the presence of cryptic prophages, since all tested phages grew very well with restored O16-type O-antigen expression. All variants of *E. coli* K-12 used for phenotyping carried an F-plasmid without *pifA* defense gene (unless that was to be tested) and pBR322_ΔP*tet* as empty vector control or plasmids expressing different antiviral defenses (see *Bacterial strains* above and S3 Table for a list). Note that the phenotyping of phages sensitive to the presence of cryptic prophages enabled us to use the isogenic host with prophages as a test condition for efficiency of plating but prevented us from testing phage sensitivity to the EcoKI type I RM system because the *E. coli* K-12 BW251139 Δ9CP strain has active EcoKI methylation (see *Bacterial strains* above and the genotype in S3 Table). It is therefore highlighted at all graphs whether ΔRM (EcoKI tested, cryptic prophage sensitivity not tested) or Δ9CP (EcoKI not tested, cryptic prophage sensitivity tested) have been used as hosts. To test for the effect of surface glycans on phage infectivity, we used host variants expressing no O-antigen but—unlike in our previous work [10]—no mutants with deeper truncations in the LPS core because almost all phages completely depended on O-antigen expression. The results with an *E. coli* K-12 mutant lacking NGR expression due to the inactivation of *nfrA* are merely shown in S1A Fig since evidently no new phage isolate used this glycan for host recognition.

For these quantitative experiments, we prepared top agars on regular LB agar plates by overlaying them with a soft agar composed of LB agar containing only 0.5% agar and additionally 20 mM $MgSO_4$ as well 5 mM $CaCl_2$ supplemented with a suitable bacterial inoculum. We used 3 ml of this soft agar with 100 μl of bacterial overnight culture as inoculum for regular round Petri dishes (ca. 9.4 cm diameter) and 10 ml of top agar supplemented with 200 μl of bacterial overnight culture for larger square Petri dishes (ca. 12 cm × 12 cm). EOP experiments were performed with serial dilutions of phage stocks grown on the respective reference hosts prepared in sterile phosphate-buffered saline (PBS). For each experiment, 2.5 μl of all serial dilutions were spotted on the top agar plates and dried into the top agar before incubation at 37 °C for at least 24 hours. We then recorded plaque formation repeatedly throughout this time to accurately quantify the infectivity of fast- and slow-growing phages. Finally, we determined the EOP of each phage on each host as the ratio of plaques obtained on the experimental host over the number of plaques obtained on the respective reference strain [177]. Plaque counts and the calculation of EOP of all replicates of all experiments reported in this study (Figs 5D, 6G, 7D, 8G, 9E, 10E, 13B, S1A and S1B) as well as their summary statistics are compiled in S1 Data.

In cases when visual inspection could not unambiguously record plaque formation, we determined the EOP to be below the detection limit of $10^{-6}$ even if significant lysis from without was apparent (i.e., lysis zones caused by bacterial cell death without significant phage replication simply from the death of infected cells [123]). In cases where a phage/ host pair was on the edge between strong lysis from without and very poor plaque formation, we cautiously recorded the result as an EOP below detection limit.

## Qualitative top agar assays

We also determined the lysis host range of all tested phages on diverse *E. coli* strains (including K-12 ΔRM or Δ9CP with and without restored O-antigen) as well as *Salmonella* Typhimurium, i.e., the range of host strains on which a certain phage caused lysis zones irrespective of whether these were accompanied by plaque formation or not (Fig 13A). These experiments were performed as qualitative top agar assays, as described in our previous work [10]. Briefly, top agar plates of each host strain were prepared as described for efficiency-of-plating assays. After solidification, we spotted each 2.5 μl of undiluted high-titer stocks of all tested phage isolates (typically >$10^9$ pfu/ml) onto the top agar plates. When the spots had dried in, we incubated the plates at 37 °C for at least 24 hours before visual inspection. The appearance of a lysis zone in more than half of all replicates for a phage–host pair meant that the host was counted as part of the lysis host range for this phage. In all such cases, we subsequently quantified phage infectivity in efficiency-of-plating assays (see above).

## Bacteriophage genome sequencing and assembly

Genomic DNA of phage isolates was prepared from culture lysates using the Norgen Biotek Phage DNA Isolation Kit and sequenced at SeqCenter (https://www.seqcenter.com/) or SeqCoast Genomics (https://seqcoast.com/) using Illumina technology. The short reads (151 bp) were trimmed and assembled using the Geneious Assembler implemented in Geneious Prime 2024.0.2 with a coverage of typically 50–100× (see S2 Table for each phage). Circular contigs (indicating complete assembly of a viral chromosome due to the fusion of characteristic repeats at the genome ends [178]) were typically obtained easily using the "Low Sensitivity/ Fast" setting with ca. 100′000 reads as input for the assembly. If necessary, ambiguities or incomplete assemblies were resolved by PCR amplification of the high-fidelity polymerase Phusion (NEB) followed by Sanger Sequencing. Sequences were linearized at the 5′ end either

to the first nucleotide of the small or large terminase subunit gene or to the first position of the operon containing the small or large terminase subunit gene.

### Initial bacteriophage genome annotation

The genomes of new phage isolates were annotated successively in several steps. To start, we used Pharokka to generate an automated, preliminary annotation of the genes in all phage genomes, which included tRNA prediction using tRNAscan-SE [179,180]. Subsequently, the annotations were refined manually based on identified orthologs in whole-genome alignments to different related phages generated using MAFFT v7.490 [181] implemented in Geneious Prime 2024.0.2. In parallel, we used the InterPro 101.0 protein domain signature database to maximize the functional information accessible from the primary sequence of phage proteins [114].

### Curation of genome annotations using remote homologies and structure prediction

For the curation of phage genome annotations, we developed a computational pipeline called GAPS (Gene Annotation Pipeline based on sequence and structure Similarity). Briefly, this pipeline employs a combination of sequence and structure analyses, database searches, interpretation, annotation, and visualization routines to infer functional information for a given set of genes, of any organism, in an automated way. For the current study, phage gene sequences were fed to GAPS as features in a.gb file and translated to protein. For each protein sequence: (i) the protein and domain families were determined by searching the Pfam, NCBI_CD, CATH_S40 and PHROGS databases with HHblits [182–187]; (ii) identical sequences were found by searching the PDB, SwissProt and RefSeq databases [188–190]; (iii) similar sequences were found by searching the UniRef30 and PDB70 databases with HHblits [186–189]; (iv) the protein structures were predicted with AlphaFold2 [191]; (v) similar structures were found by searching the PDB, AlphaFold-Proteome and AlphaFold-UniProt50 databases with Foldseek [188,191,192]. The results were parsed and integrated programmatically for presentation in an interactive report file that summarizes the available information for all genes of the input genome in one combined overview page. Detailed results of all individual genes are presented in subpages. An exemplary report for *Jilinvirus* HeidiAbel (Bas97) is included in this publication as S2 Data and the added value of GAPS for the functional annotation of its genes is highlighted in S13 Fig. GAPS is written in Python 3.8 and is open-source. The code used to run GAPS for generating the data presented in this manuscript is available on GitHub (https://github.com/hiller-lab/gaps) and on Zenodo (https://www.doi.org/10.5281/zenodo.14277981) in the form of a Jupyter notebook.

### Bacteriophage naming and taxonomy

Phage isolates were classified according to applicable rules of the International Committee on the Taxonomy of Viruses (ICTV) [68,193] largely as described in our previous work BASEL [10] (see S2 Table). Briefly, phages were first roughly assigned to viral family and, if possible, genus based on whole-genome BLASTN searches against the nonredundant nucleotide collection database [https://blast.ncbi.nlm.nih.gov/Blast.cgi; see also reference 194]. Subsequently, we unraveled the evolutionary relationships between our new isolates and reference sequences from NCBI GenBank across the diversity of relevant viral groups by calculating phylogenies based on either whole-genome alignments or alignments of conserved marker genes (see details in the dedicated section below). We then compared the clusters of viral sequences observed in these phylogenies to phage taxonomy as established by the ICTV and the relevant

literature [68,195]. For classification on the species level, whole-genome alignments of new phage isolates with close relatives were generated as described below, and the nucleotide sequence identity was determined as the fraction of aligned genome sequence multiplied by the nucleotide identity of aligned segments. Phage isolates were classified as the same species as previously described relatives if their genomes showed >95% nucleotide sequence identity [196].

We named most phages in honor of scientists or historically significant persons linked to the city of Basel, Switzerland, where most of the phages had been isolated. To overcome the inherent gender imbalance due to historical biases, we included diverse female personalities from across the whole of Switzerland. Phages of *Skatevirus* and *Nonanavirus* genera were given names composed of common female first names in Switzerland in 1971 (when Swiss women obtained voting rights in federal elections) and "Muster" as surname (the German word for paradigm or example).

## Sequence alignments and phylogenetic analyses

Sequence alignments of bacteriophage genomes, specific loci, or certain genes/ proteins were generated using MAFFT v7.490 implemented in Geneious Prime 2024.0.2 [181]. Occasionally, we supplemented poor or missing annotations in these sequences using the ORF finder tool of Geneious Prime 2024.0.2 guided by orthologous loci. Alignments were typically constructed using default settings of MAFFT with the fast FFT-NS-2 algorithm and 200PAM/k=2 or BLOSUM62 scoring matrices for nucleotide and amino acid sequences, respectively. Subsequently, we curated the alignments manually and masked nonhomologous parts for exclusion during phylogenetic analyses.

Maximum-likelihood phylogenies were calculated using PhyML 3.3.20180621 implemented in Geneious Prime 2024.0.2 from sequence alignments of orthologous genes/ proteins or orthologous sequence stretches of different bacteriophage genomes [197]. We used the HYK85 and LG substitution models for nucleotide and amino acid sequences, respectively, and always calculated 100 bootstraps. Phylogenetic relationships between phage genomes were inferred from whole-genome alignments (whenever possible) or from concatenated sequence alignments of conserved core genes on nucleotide or amino acid level as indicated for each phylogeny. Relevant reference sequences and controls were selected to represent the diversity of taxonomic units or protein families and typically identified by BLAST searches against the non-redundant nucleotide collection database [https://blast.ncbi.nlm.nih.gov/Blast.cgi; see also reference [194]. When available, well-studied model phages such as P22 and their sequences were always included to provide intuitive reference points for data interpretation. The tree files and underlying sequence alignments in NEXUS format are included in this manuscript as S3 Data file.

## Structure predictions and analyses of receptor-binding proteins

Structure predictions were performed using AlphaFold-Multimer on the Cosmic² platform or on the AlphaFold3 webserver [198–200]. All predictions presented high internal confidence scores, which are presented in S7 Table. Illustrations were generated using the PyMOL Molecular Graphics System (Version 2.4.1, Schrodinger LLC). The Dali server was used to identify structural homologs in the Protein Data Bank (PDB) [201].

## Morphological analyses by transmission electron microscopy

We analyzed the virion morphology of representative isolates from the different taxonomic groups covered in this manuscript by negative-stain transmission electron microscopy [202].

Briefly, we applied 5 μl drops of high-titer phage lysates to 400 mesh carbon-coated grids that had been rendered hydrophilic using a glow-discharger at low vacuum conditions. Subsequently, virions were stained on 5 μl drops of 2% (w/v) uranyl acetate and examined using an FEI Tecnai G2 Spirit transmission electron microscope (FEI Company) at 80-kV accelerating voltage, a Hitachi HT7700 transmission electron microscope (Hitachi) at 100 kV, or an FEI Talos F200C transmission electron microscope (FEI Company) at 200 kV. Images were recorded with a side-mounted Olympus Veleta CCD camera 4k (Tecnai G2 Spirit), an AMT XR81B Peltier cooled CCD camera (8 M pixel; Hitachi HT7700), or a Thermo Fisher Scientific Ceta 16M CCD camera (Talos F200C) using EMSIS RADIUS software at a nominal magnification of typically 150′000×.

## Quantification and statistical analysis

Data sets were analyzed quantitatively by calculating mean and standard deviation of at least three independent biological replicates per experiment. Detailed information about replicates and statistical analyses is provided in the figure legends. In addition, we determined the probability for all efficiency-of-plating values that they are significantly different from the *E. coli* K-12 vector control using Student *t* test (two-sample unequal variance/ heteroscedastic) with a result considered significant if p<0.01 (included in S1 Data).

## Supporting information

**S1 Table. Isolation of O16-dependent and K12-prophage-inhibited phages from sewage and environmental samples.**
(XLSX)

**S2 Table. List of all phages used in this study.** This table presents the taxonomic classification, isolation/ source, host receptors, and genomic features of all phages characterized in this study. The original BASEL phages from our previous work (Bas01-Bas69; [10]) and classical reference phages are included for completeness. Predictions of anti-defense proteins encoded by all BASEL phages (using AntiDefenseFinder [128]) are included on a second sheet for a better overview. The genomes of *Lederbergvirus* phages Huey, Dewey, and Louie are available in NCBI GenBank with accession numbers PQ850600, PQ850609, and PQ850616.
(XLSX)

**S3 Table. List of all bacterial strains used in this study.** This table presents all bacterial strains that have been used in this work. The abbreviations in the selection column indicate the drug and its concentration that were used. Amp = ampicillin, Cam = chloramphenicol, Kan = kanamycin, Zeo = zeocin; 25/ 50/ 100 refer to 25 μg/ml, 50 μg/ml, and 100 = 100 μg/ml, respectively.
(DOCX)

**S4 Table. List of all oligonucleotide primers used in this study.**
(DOCX)

**S5 Table. List of all plasmids used in this study.** Abbreviations in the selection column highlight the drug and its concentration that were used to select for plasmid maintenance. Amp = ampicillin, Cam = chloramphenicol, Kan = kanamycin; 25/50/100 refer to 25 μg/ml, 50 μg/ml, and 100 = 100 μg/ml, respectively.
(DOCX)

**S6 Table. Construction of all plasmids generated in this study.**
(XLSX)

**S7 Table. Confidence scores of AlphaFold-generated models.** Confidence scores of AlphaFold-generated models. Confidence per residue is calculated as a predicted Local Distance Difference Test score (0–100), with an average of all residues within the models provided below. A pLDDT ≥ 90 have very high model confidence, residues with 90 > pLDDT ≥ 70 are classified as confident, while residues with 70 > pLDDT > 50 have low confidence, and residues with pLDDT <50 correspond to very low confidence. Interface pTM scores ("iptm+ptm") are a measure of predicted protein multimer accuracy by AlphaFold-Multimer using the with confidence scored 0–1 for the complete model. All structures were predicted using the COSMIC2 platform, except ManiWeber Gp32 (*) solved using the AlphaFold 3 Server. (XLSX)

**S1 Data. Raw data and calculations of all efficiency-of-plating (EOP) assays and qualitative top agar experiments.** (XLSX)

**S2 Data. GAPS report for the HeidiAbel genome.** (ZIP)

**S3 Data. Tree files with underlying sequence alignments of all phylogenies.** (ZIP)

**S1 Fig. Supplemental data for Fig 4.** (A) Phages that had shown significant growth on *E. coli* K-12 with restored O-antigen in our previous work [10] were now phenotypically characterized in the presence of this glycan. The results of these quantitative phenotyping experiments regarding sensitivity to altered surface glycans and bacterial immunity systems are presented as efficiency of plating (EOP). The small note of "ΔRM" indicates that this experiment has been performed using *E. coli* K-12 ΔRM *wbbL*(+) as host (i.e., with restored O-antigen expression). Data points and error bars represent average and standard deviation of at least three independent experiments. Raw data and calculations are available in S1 Data. (B) The results of quantitative phenotyping experiments with all newly presented BASEL phages infecting an *nfrA* knockout (specifically lacking the NGR glycan [17]) are presented as efficiency of plating (EOP). This strain had been transformed with plasmid pAS001 to restore O16-type O-antigen expression (see Materials and methods). Data points and error bars represent average and standard deviation of at least three independent experiments. Raw data and calculations are available in S1 Data. It is evident that no newly included BASEL phage depends partially or completely on the NGR glycan for infection of *E. coli* K-12. Phages sensitive to the presence of cryptic prophages in the K-12 genomic background could not be tested. However, since these phages all carry tailspikes or tail fibers targeting O16-type O-antigen as receptor-binding proteins (Figs 11 and 12) it seems highly unlikely that they would be affected by the absence of NGR. (TIF)

**S2 Fig. Supplemental data for Fig 6.** (A) The illustration compares the tail fiber and tail-spike loci of diverse small siphoviruses from the original BASEL collection and the current study and porin-targeting receptor-binding proteins (RBPs) are annotated based on the *bona fide* central tail fiber caps identified previously [10]. It is apparent that the single RBPs of the three exemplary O16-dependent phages SonjaBuckley (*Kagunavirus*, Bas74), SandraMuster (*Skatevirus*, Bas78), and KarinMuster (*Nonanavirus*, Bas82) are encoded at the same locus as the porin-targeting RBPs for terminal receptor recognition in the other genomes. For comparison, we also included two small siphoviruses targeting *Klebsiella* capsules that had been classified in previous work as belonging to *Nonanavirus* (see Fig 6E) and *Roufvirus* (see Fig

6D) [49]. Despite the annotation as "tail fiber," both viruses encode tailspikes directly downstream of the orthologs of the J-like central tail fiber proteins [49]. (B) Comparison of the *bona fide* central tail fiber RBP loci of all *Dhillonvirus* isolates in the BASEL collection as well as PTXU06 for comparison [10,58]. The association of different variants of this locus with porin specificity is color-coded as in our previous work – all new isolates from the current study are predicted to target LptD (red) as terminal receptor [10].
(TIF)

**S3 Fig. Supplemental data for Fig 7.** The conserved locus encoding dCTP deaminase, nucleotidyltransferase, and DNA replication genes of all included *Gordonclarkvirinae* isolates is shown in a sequence alignment (see Materials and methods). Colors in the sequence identity graph above the alignment indicate the sequence identity at each individual position with green representing 100% identity, greenish-brown 30%−99% identity, and red <30% identity. Besides homing endonuclease insertions, the locus is fully syntenic across these isolates (and many other *Gordonclarkvirinae*).
(TIF)

**S4 Fig. Supplemental data for Fig 8.** The genomes of our new isolate HelliStehle (Bas95) and *Xuquatrovirus* phages Pondi (GenBank accession OP136151.1) and PTXU04 (NCBI GenBank accession NC_048193.1) are shown in a sequence alignment with the putative small terminase subunit as starting point (see Materials and methods). Colors in the sequence identity graph above the alignment indicate the sequence identity at each individual position with green representing 100% identity, greenish-brown 30%−99% identity, and red <30% identity. The pairwise sequence identity of prototypic phage PTXU04 to fellow *Xuquatrovirus* Pondi is 83.8% and to HelliStehle (Bas95) merely 58.3%.
(TIF)

**S5 Fig. Supplemental data for Fig 9.** (A) Serial dilutions of *Jilinvirus* HeidiAbel (from left to right) were spotted on a soft agar of *E. coli* K-12 ΔRM *wbbL*(+) and grown overnight at 37°C before imaging. The lysis zones and plaques show no classical signs of lysogen formation such as turbidity or marked growth of (self-immune) lysogens in the middle [108] – compare, e.g., the growth of temperate *Lederbergvirus* phages Huey, Dewey, and Louie in S6B Fig or of phage Kap1 on a host enabling lysogeny and a host without its preferred integration site in Fig 3 of Pick and colleagues [205]. Instead, upon prolonged incubation merely an abundance of dotted individual colonies is found that we typically observe for phages requiring the LPS O-antigen as host receptor, which is a huge mutational target [206]. (B, C) The illustrations show sequence alignments of the integrase locus of *Iiscvirinae* phages (B) and – encoded slightly downstream – their cI-like repressor locus (C; see Materials and methods). Both genes of interest are highlighted in blue. Colors in the sequence identity graphs above the alignments indicate the sequence identity at each individual position with green representing 100% identity, greenish-brown 30%−99% identity, and red <30% identity. The alignments show that only the two *Pseudomonas* phages (top) and *Salmonella* phage PT1 as well as our isolate HeidiAbel encode seemingly intact integrase and cI-like repressor genes. These observations are further discussed in the main text.
(TIF)

**S6 Fig. Supplemental data for Fig 10.** (A) The illustration shows a Maximum-likelihood phylogeny (left) and a sequence alignment (right) of Gp1.2 orthologs in all tested *Autographiviridae* phages in the complete BASEL collection. Experimental details are explained in Materials and methods. Resistance to the PifA defense system (see Fig 10E and our previous work [10]) correlates with one specific clade of Gp1.2 homologs (the middle one). (B) Plaque

morphology of our four *Lederbergvirus* isolates infecting *E. coli* K-12 with restored O16-type O-antigen streaked on *E. coli* K-12 ΔRM *wbbL*(+). While CurroJimenez (Bas106) forms clear plaques of variable size, the other three isolates form highly turbid plaques that are indicative of lysogen formation.
(TIF)

**S7 Fig. Supplemental data for Figs 11 and 12.** AlphaFold-Multimer [198] predicted structures of additional tail fibers from selected BASEL phages with a single receptor-binding domain (RBD). Tail fibers of *Teseptimavirus* phages MeretOppenheim (Bas102) and EmilieLinder (Bas103) are formed by a homotrimer of Gp58 or Gp59, respectively. Note that the tail fiber of MeretOppenheim contains a C-terminal intramolecular chaperone (IMC) domain. Conversely, EmilieLinder encodes a separate Mu-like tail fiber assembly protein (Tfa, Gp60; green) downstream of the Gp59 tail fiber that are predicted to form a hexameric complex as observed for the tail fiber of phage Mu [207]. The homotrimeric tail fibers of *Xuquatrovirus* HelliStehle (Bas95) and *Jilinvirus* HeidiAbel (Bas97) feature comparably short fiber stalks. HeidiAbel also encodes a Tfa protein (Gp16; green) that is predicted to form a complex with the tail fiber similar to the complex predicted for EmilieLinder. In addition, HeidiAbel also encodes a separate tail fiber with a distal tip that is similar, but truncated, to the T4 Gp37 long-tail fiber and the Gp32 tail fiber of *Wifcevirus* ManiWeber [37] (see Fig 12C). Based on the size and genetic arrangement, we speculate that the Gp15 fiber of HeidiAbel serves as a lateral tail fiber for initial host recognition while Gp17 forms a short-tail fiber for irreversible adsorption triggering viral genome injection (analogous to the much larger T-even myoviruses [10,73,74]). In all predicted structures, N-terminal anchor domains and fiber shafts are colored blue with the RBDs colored red. A ~5 nm vertical scale bar is provided for reference and viral morphotypes are annotated by phage icons analogous to Fig 11A and 11B.
(TIF)

**S8 Fig. Supplemental data for Figs 11 and 12.** The illustration shows a sequence alignment of the tailspike locus of different *Gamaleyavirus* phages (compare Fig 8E; see Materials and methods). Colors in the sequence identity graph above the alignment indicate the sequence identity at each individual position with green representing 100% identity, greenish brown 30%–99% identity, and red <30% identity. The alignment shows that the most 5′ parts of the two tailspike loci – encoding the N-terminal ends of the (proximal subunits of the) tailspikes – are highly conserved, likely encoding conserved structural elements that anchor them at the virion. The receptor-binding domains of all tailspikes vary greatly. These observations are further discussed in the main text.
(TIF)

**S9 Fig. Supplemental data for Figs 11 and 12.** (A) The illustration shows an amino acid sequence alignment of the short-tail fiber proteins of all *Gordonclarkvirinae* included in Fig 7C in the same order (see Materials and methods). It is apparent that short-tail fiber proteins are highly conserved across the *Gordonclarkvirinae* subfamily with only minor differences. Across all sequences there are 163 of 267 fully identical sites (60.8%). (B) Amino acid sequence alignment of the conserved tail fiber of *Wifcevirus* phages encoded with their tail fiber locus (Gp32 of ManiWeber/ Bas99, see Fig 12C). It is clear from the alignment that all orthologs are very similar in amino acid sequence, suggesting a conserved and invariable function. (C) Amino acid sequence alignment of the T-shaped tail fibers of *Wifcevirus* phages (see Figs 11B, 12B and S10D). The alignment shows that the N-terminus of all orthologs is highly conserved while the two receptor-binding domains (RBDs) as well as – to lesser extent – the intramolecular chaperone of the T5-like RBD2 vary a lot between sequences. In (A) to (C), colors in the

sequence identity graph above the alignment indicate the sequence identity at each individual position with green representing 100% identity, greenish brown 30%–99% identity, and red <30% identity.
(TIF)

**S10 Fig. Supplemental data for Figs 11 and 12.** (A) The AlphaFold-Multimer [198] predicted structure of the T-shaped tail fiber of IrisVonRoten (Gp24) was colored to highlight the domain boundaries of this unique dual RBD tail fiber. (B) The same structure as in (A) was colored by predicted local distance difference test (pLDDT) confidence scores as generated by AlphaFold to demonstrate the high level of confidence (overall average = 72.6) in the predicted T-shaped tail fiber structure. (C) The DALI server [201] identified structural similarities between the Gp12-like RBD1 with crystal structures of the distal knob domain (D10) of the T4 long-tail fiber Gp37 [118] (Z-score 8.8, RMSD 3.2, 100 residues) and unsurprisingly with the Gp12 short-tail fiber distal tip itself [118] (Z-score 6.2, RMSD 4.1, 127 residues) as well as similarity between the T5-like RBD2 with the distal binding tip and IMC domain of the phage T5 lateral tail fiber [119] (Z-score 6.3, RMSD 4.8, 179 residues). (D) Maximum-likelihood phylogeny of T5-like receptor-binding domains of T-shaped tail fibers together with their intramolecular chaperone domain as identified by structure prediction (see Materials and methods). The phylogeny was midpoint-rooted between a clade of distant *Gordonclarkvirinae* tail fibers at the top (blue) together with two other sequences and all other RBDs mostly belonging to phages presented in this study (orange). Note how RBDs of *Wifcevirus*, *Gordonclarkvirinae*, and *Tequintavirus* isolates are scattered over the phylogeny in small clusters, suggesting frequent horizontal gene transfer as often observed for viral receptor-binding domains [10,33,115]. Phage AidaStucki (Bas93, *Gordonclarkvirinae*) was not included due to a large truncation in its intramolecular chaperone domain.
(TIF)

**S11 Fig. Supplemental data for Fig 14.** The same serial dilutions as in Fig 14B were spotted on variants of *E. coli* K-12 BW25113 without restored O-antigen expression (left) and with either deletion of the GtrS glucosyltransferase (middle) or the nine cryptic prophages including CPS-53 (right). In difference to the results shown in Fig 14B with restored O-antigen expression, the three *Dhillonvirus* phages Bas83–85 readily infect all of the strains lacking O-antigen expression.
(TIF)

**S12 Fig. Supplemental data for Materials and methods.** The same serial dilutions as in Fig 1B (for (A)) and 2B (for (B)) were spotted on the *E. coli* K-12 ΔRM strain with partially restored O16-type O-antigen due to ectopic expression of *wbbL* from plasmid pAS001 (see Materials and methods). Most but not all phages inhibited by a full O16-type O-antigen barrier (green) show robust growth on this host (compare Fig 1B) while those that can bypass it (brown; compare Fig 1B) or that depend on it (blue; compare Fig 2B) grow very well. Notably, a weak inhibition of phage growth had already been observed previously for intermediate restoration of the *E. coli* K-12 O16-type O-antigen [30].
(TIF)

**S13 Fig. Supplemental data for Materials and methods.** The illustration shows the genome of phage HeidiAbel (Bas97; see Fig 8F) with annotations highlighted in different colors. Blue annotations had been predicted by Pharokka in a similar functional context but could be improved by GAPS while red annotations had only been predicted as hypothetical proteins by Pharokka [179]. The genes highlighted in gray had not been predicted by Pharokka in this setup but were inferred by manual analysis of a whole-genome alignment between HeidiAbel

and related phages (see Materials and methods). A tRNA is highlighted in pink. The GAPS output for HeidiAbel based on the Pharokka-annotated genome as input is included as S2 Data.
(TIF)

## Acknowledgments

The authors are grateful to Dr. Julie Sollier and Valerie Oriet for valuable input and critical reading of the manuscript. We thank high school students participating in the Biozentrum Summer Science Academy 2022 for assistance with the isolation of multiple bacteriophages described in this study (S2 Table). Carmen Saez Garcia Wanzenried is acknowledged for her invaluable help with replicating the BASEL phages and for naming phage CurroJimenez (Bas106). The authors are indebted to the ScopeM facility at ETH Zürich (Switzerland) and the team at the BioEM lab of Biozentrum, University of Basel (Switzerland) for their support with transmission electron microscopy. We further thank ARA Basel, ARA Canius (Lenzerheide), and ARA Triengen for samples of sewage plant inflow. The authors are grateful to David Schwegler for pioneering work on the annotation of phage genomes using remote homologies and structure prediction with GAPS. Dr. Toomas Mets (University of Tartu, Estonia) is acknowledged for the initial discovery that partial restoration of O16-type O-antigen expression with pAS001 can still inhibit the growth of some phages. The authors are grateful to Dr. Jiemin Du, Dr. Samuel Kilcher, and Prof. Dr. Martin Loessner (ETH Zürich, Switzerland) for helpful discussions about the impact of bacterial O-antigen on phage host range. Dr. Artem Isaev is acknowledged for a helpful comment on anti-phage defense systems in cryptic prophages. We are grateful to the Coli Genetic Stock Center (CGSC) for the *E. coli* K-12 EMG2 strain and to the German Collection of Microorganisms and Cell Cultures (DSMZ) for phage lambda PaPa.

## Author contributions

**Conceptualization:** Dorentina Humolli, Damien Piel, Enea Maffei, Elia Agustoni, Adrian Egli, Sebastian Hiller, Matthew Dunne, Alexander Harms.

**Data curation:** Dorentina Humolli, Damien Piel, Enea Maffei, Yannik Heyer, Elia Agustoni, Fabienne Estermann, Aline Cuénod, Carola Alampi, Mohamed Chami, Adrian Egli, Sebastian Hiller, Matthew Dunne, Alexander Harms.

**Formal analysis:** Dorentina Humolli, Damien Piel, Enea Maffei, Yannik Heyer, Elia Agustoni, Fabienne Estermann, Aline Cuénod, Carola Alampi, Mohamed Chami, Sebastian Hiller, Matthew Dunne, Alexander Harms.

**Funding acquisition:** Alexander Harms.

**Investigation:** Dorentina Humolli, Damien Piel, Enea Maffei, Yannik Heyer, Elia Agustoni, Aisylu Shaidullina, Luc Willi, Patrick Imwinkelried, Carola Alampi, Mohamed Chami, Matthew Dunne, Alexander Harms.

**Methodology:** Dorentina Humolli, Damien Piel, Enea Maffei, Yannik Heyer, Elia Agustoni, Aisylu Shaidullina, Luc Willi, Patrick Imwinkelried, Aline Cuénod, Dominik P. Buser, Carola Alampi, Mohamed Chami, Matthew Dunne, Alexander Harms.

**Project administration:** Fabienne Estermann, Dominik P. Buser, Adrian Egli, Sebastian Hiller, Alexander Harms.

**Resources:** Dorentina Humolli, Damien Piel, Enea Maffei, Yannik Heyer, Elia Agustoni, Fabienne Estermann, Aline Cuénod, Dominik P. Buser, Carola Alampi, Mohamed Chami, Adrian Egli, Sebastian Hiller, Matthew Dunne, Alexander Harms.

**Software:** Dorentina Humolli, Damien Piel, Enea Maffei, Elia Agustoni, Fabienne Estermann, Sebastian Hiller, Matthew Dunne.

**Supervision:** Enea Maffei, Fabienne Estermann, Dominik P. Buser, Mohamed Chami, Adrian Egli, Sebastian Hiller, Alexander Harms.

**Validation:** Dorentina Humolli, Damien Piel, Enea Maffei, Yannik Heyer, Elia Agustoni, Aisylu Shaidullina, Luc Willi, Patrick Imwinkelried, Sebastian Hiller, Matthew Dunne, Alexander Harms.

**Visualization:** Dorentina Humolli, Damien Piel, Enea Maffei, Elia Agustoni, Fabienne Estermann, Sebastian Hiller, Matthew Dunne, Alexander Harms.

**Writing – original draft:** Alexander Harms.

**Writing – review & editing:** Dorentina Humolli, Damien Piel, Enea Maffei, Yannik Heyer, Elia Agustoni, Fabienne Estermann, Aline Cuénod, Carola Alampi, Mohamed Chami, Adrian Egli, Sebastian Hiller, Matthew Dunne, Alexander Harms.

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
