## [Editor Report · Decision Letter 0]

25 Sep 2024

Dear Dr Harms, 

Thank you for submitting your manuscript entitled "Unlocking hidden phage diversity to complete the BASEL collection for systematic exploration of phage-host interactions" for consideration as a Methods and Resources by PLOS Biology. I would like to apologize for the delay in coming back with a response.

Your manuscript has now been evaluated by the PLOS Biology editorial staff, as well as by an academic editor with relevant expertise, and I am writing to let you know that we would like to send your submission out for external peer review.

Once your full submission is complete, your paper will undergo a series of checks in preparation for peer review. After your manuscript has passed the checks it will be sent out for review. To provide the metadata for your submission, please Login to Editorial Manager (https://www.editorialmanager.com/pbiology) within two working days, i.e. by Sep 27 2024 11:59PM.

Kind regards,

Melissa

Melissa Vazquez Hernandez, Ph.D.

Associate Editor

PLOS Biology

---

## [Decision Letter · Decision Letter 1]

22 Nov 2024

Dear Alex,

Thank you for your patience while your manuscript "Unlocking hidden phage diversity to complete the BASEL collection for systematic exploration of phage-host interactions" went through peer-review at PLOS Biology. Your manuscript has now been evaluated by the PLOS Biology editors, an Academic Editor with relevant expertise, and by two independent reviewers. I would really like to apologize for the extremely long delayed in coming back to you with a decision. 

In light of the reviews, which you will find at the end of this email, we are pleased to offer you the opportunity to address the comments and suggestions from the reviewers in a revision that we anticipate should not take you very long. We would like to encourage that you follow all the suggestions made. We will then assess your revised manuscript and your response to the reviewers' comments with our Academic Editor aiming to avoid further rounds of peer-review, although might need to consult with the reviewers, depending on the nature of the revisions.

Additionally, to move forward faster, we would like to already ask you to address the following editorial requests:

a) We routinely suggest changes to titles to ensure maximum accessibility for a broad, non-specialist readership, and to ensure they reflect the contents of the paper. Please ensure you change both the manuscript file and the online submission system, as they need to match for final acceptance:

"BASEL phage collection expansion unlocks hidden diversity for systematic exploration of phage-host interactions"

b) Thank you for providing all numerical values for the figures and for citing the location in the figure legends. I took a look at the Data_S1 file, and it is a bit difficult to know to which experiment the numbers belong to. If possible, you could separate them in individual sheets depending on the figure. Thank you.

c) Please provide the tree files for Figures 5C, 6DEF, 7C, 8DEF, 9B, 10B, 11AB, S6, S10

d) Many thanks for providing the underlying code in GitHub. However, because Github depositions can be readily changed or deleted, please make a permanent DOI’d copy (e.g. in Zenodo) and provide this URL in the manuscript and Data Availability Statement.

While the revisions should not take longer than a month, due to Christmas holidays we have extended the deadline to January 6. Let me know if you might need more time. 

**IMPORTANT - SUBMITTING YOUR REVISION**

*Resubmission Checklist*

*Published Peer Review*

*PLOS Data Policy*

*Blot and Gel Data Policy*

Sincerely,

Melissa

Melissa Vazquez Hernandez, Ph.D.

Associate Editor

PLOS Biology

REVIEWERS' COMMENTS:

Reviewer #1: 

The manuscript by Humolli, Maffei, Heyer, Agustoni, and colleagues presents an extension of the BASEL collection, adding phages that target O-antigen or are affected by defence systems encoded by cryptic prophages in E. coli K12. This well-characterised collection of E. coli phages is a fantastic tool for phage-host interaction research, and this new addition to the collection is certainly worthy of publication. I have only a few minor comments for the authors to address prior to publication.

1) At the end of the Introduction, I suggest the authors provide a brief overview of the updated BASEL collection, including the total number of phages and phage groups it now comprises. 

2) On page 14, line 221, please clarify that the phage sensitivity observed applies only to the defence systems tested, as these phages may still be resistant to other defences not present in K12. 

3) For phage groups (e.g. Gordonclarkvirinae, Wifcevirus, RudolfBernhard) shown to resist RM systems despite lacking known phage DNA modification pathways, have the authors checked for specific anti-RM proteins (e.g. Ocr)? DefenseFinder may be useful for this purpose. 

4) Page 17, line 269 "the Gordonclarkvirinae are also sensitive to the Fun/Z defence system, though a detectable decrease in the efficiency of plating was only observed for the Kuravirus genus". Do the authors mean that phages without reduced EOP were still sensitive to Fun/Z? If this sensitivity was observed in plaque size reduction, please include these data. 

5) Statistical analysis is missing from all EOP assays shown in the figures.

6) For panel A of figures describing phage groups, can the authors indicate when the model is hypothetical?

7) In Figure 1A, the colours representing the outer LPS core and O-antigen subunits are difficult to distinguish. Could the authors adjust this? Additionally, I recommend writing out the full names for "LPS" and "NGR" in the legend. 

8) In Figure 2, the nomenclature for E. coli strains varies between panels (e.g. K12 O16(+) in A and E. coli K12 ΔRM wbbL(+) in B). Please use consistent strain names throughout figures and the manuscript. 

9) In Figure 4C, the "RexAB" text is difficult to read against the dark red background. Consider using white font or a lighter red background.

10) For Figure 8, I suggest adding a TEM image of Xuquatrovirus and Schitoviridae to maintain consistency in showing a representative image for each phage group. 

11) Figure 13, panel A is easier to interpret than panel B. Could the results of panel B be represented as in panel A? While showing efficiency of plating is essential, a heatmap could simplify visualisation. 

12) On page 51, line 905, there is a small typo: "manuscript can is available" should be "manuscript is available".

13) In Figure S1, please indicate in panel B that the strain is ΔNGR glycan to improve clarity. 

14) In Figure S2B, include only the porin names targeted by the RBPs shown. 

Reviewer #2: 

Dr. Harms and colleagues expand their earlier BASEL collection of E. coli phages with phages that require O-antigen as a receptor. E. coli phage biology has been historically biased towards phages that infect rough (O-antigen lacking) strains as the widely used K-12 strain has lost its ability to produce O-antigen. By restoring this ability, the authors have filled in some of the gaps in diversity of E. coli phages that are available for study to scientists in the field. This is an extremely valuable resource, as this entire category of phages has been neglected for decades. Especially since almost all wild/pathogenic strains of E. coli produce O-antigen, these phages have the potential to inform research in a critical aspect of phage biology that is a necessary pre-requisite of phage therapy. Moreover, the careful characterization of these new phages is just as important a contribution as the actual biological materials described herein, if not more so. By exercising great care in their characterizations, the authors also make some important discoveries about the function and distribution of phage receptor binding proteins. For instance, it is surprising that non-enzymatic tail fibers are so widely represented in O-antigen dependent phages. Also, the authors' description of T-shaped tail fibers with distinct RBDs, possibly allowing for broader host range is striking, and may prove to be very useful for phage therapy design. Overall, the writing is clear and well contextualized. The paper is thorough, complete, and exquisitely supported by supplementary data. The big question of course is how much more diversity will be uncovered by isolating phages on other E. coli O-antigen serotypes. While the answer to this question is obviously beyond the scope of this paper, we are hopeful the authors will continue their valuable efforts in this line of inquiry in the future. This phage collection, along with their previous work initially describing the BASEL phages will undoubtedly be an enduring part of the canon of E. coli phage biology. 

A few questions I had while reading, in case further discussion is helpful (by no means should any of this be considered an impediment to swift publication):

1. 73-78: Were Dhillonvirus and Teseptimavirus the only groups where overlap between O-antigen dependent and O-antigen independent phages was found (comparing to the original BASEL collection)? Were there any O-antigen using (whether obligate or not) phages in the 100+ isolated (L 175-176) that also fell in the previously described groups?

2. 123-125: "but not the isogenic strain without this glycan" - was this an observation or a requirement that was imposed? What would the authors say is the relative prevalence of phages (from the 100+ isolated) that are capable of infecting strains "through" the O-antigen layer (which could be critical for phage therapy in E. coli) but which may also be isolated on strains lacking O-antigen? Essentially, phages that can use but do not strictly require the O-antigen as a receptor (e.g. T5). Is it just 2 out of 68 (Bas 07, Bas32) (L:110-112)? From L 175-176, I cannot determine how many of the 100+ new phages that are "unable to grow on regular K12" are due to exclusion by O-antigen versus prophages/defense. 

3. 120-122: Is this quantitative rarity perhaps explained by the fact that there are ~180 O-antigen serotypes in E. coli and some of the phages specific to each O-antigen serotype might not need the O-antigen as an obligate receptor, so the class of O-antigen-independent phages borrows from all other O-antigen-specific phage groups and ends up being larger than all?

4. 224-226, 243-245: It is striking that O-antigen dependent and independent Dhillonviruses do not cluster separately, and that Gordonclarkvirinae also scatter widely. I suspect the genomes are too distant to try and pin the blame on a few genes by comparative genomics, but just in case the authors have any guesses here… are the tails just evolving so quickly that they outpace the rest of the genome?

5. 229-230, 252-255: It says "unlike O-antigen dependent siphoviruses… Gordonclarkvirinae have two separate bona fide RBPs", but earlier it also says that Dhillonviruses also have two types of tail fibers with distinct RBPs?

6. 352-355: Can the authors discuss why the plaque formation is impaired without a quantitative reduction in plaque counts? It would seem that any process that inhibits successive infections after the formation of a visible plaque (which still requires many rounds of infection and re-infection) would also inhibit the first infection and therefore statistically lead to fewer plaques. Unless the progeny are somehow less diffusible?

7. Fig 11. How were the "relevant controls" chosen?

8. Fig 11B is arresting. Why is there seemingly a lot wider horizontal exchange of Gp12 in Wifcevirus than Gordonclarkvirinae? Perhaps the discussion on this point could be expanded. Perhaps this has to do with broader host range of Gordonclarkviridae with T-shaped tail fibers, which must be achieved through swapping of tailspikes for Wifcevirus? This is also a little difficult to understand because the concordance between "long" and "short" tail fibers (L 459-461) and Gp12 (L 464-466) in Gordonclarkvirinae is not clear. 

Finally, it is not clear to this reviewer whether this paper has already been reviewed and revised once. The file was labeled "Revision 1" but no commentary from a previous review was available. If this is indeed the case, I would strongly recommend publication of this paper as soon as possible. While a few cosmetic modifications may be desirable, it is unconscionable to delay publication of such a valuable community resource for nitpicking concerns of others.

---

## [Editor Report · Decision Letter 2]

11 Feb 2025

Dear Alexander,

Thank you for the submission of your revised Methods and Resources "Completing the BASEL phage collection to unlock hidden diversity for systematic exploration of phage-host interactions" for publication in PLOS Biology. On behalf of my colleagues and the Academic Editor, Jeremy J. Barr, I am pleased to say that we can in principle accept your manuscript for publication, provided you address any remaining formatting and reporting issues. These will be detailed in an email you should receive within 2-3 business days from our colleagues in the journal operations team; no action is required from you until then. Please note that we will not be able to formally accept your manuscript and schedule it for publication until you have completed any requested changes.

PRESS

Sincerely, 

Melissa

Melissa Vazquez Hernandez, Ph.D., Ph.D.

Associate Editor

PLOS Biology
